# CXCL10/CXCR3 signaling contributes to an inflammatory microenvironment and its blockade enhances progression of murine pancreatic precancerous lesions

**Veethika Pandey, Alicia Fleming-Martinez, Ligia Bastea, Heike R Doeppler, Jillian Eisenhauer, Tam Le, Brandy Edenfield, Peter Storz***

Department of Cancer Biology, Mayo Clinic Comprehensive Cancer Center, Mayo Clinic, Jacksonville, United States

**Abstract** The development of pancreatic cancer requires recruitment and activation of different macrophage populations. However, little is known about how macrophages are attracted to the pancreas after injury or an oncogenic event, and how they crosstalk with lesion cells or other cells of the lesion microenvironment. Here, we delineate the importance of CXCL10/CXCR3 signaling during the early phase of murine pancreatic cancer. We show that CXCL10 is produced by pancreatic precancerous lesion cells in response to IFNγ signaling and that inflammatory macrophages are recipients for this chemokine. CXCL10/CXCR3 signaling in macrophages mediates their chemoattraction to the pancreas, enhances their proliferation, and maintains their inflammatory identity. Blocking of CXCL10/CXCR3 signaling in vivo shifts macrophage populations to a tumor-promoting (Ym1$^+$, Fizz$^+$, Arg1$^+$) phenotype, increases fibrosis, and mediates progression of lesions, highlighting the importance of this pathway in PDA development. This is reversed when CXCL10 is overexpressed in PanIN cells.

*For correspondence:
storz.peter@mayo.edu

Competing interests: The authors declare that no competing interests exist.

## Introduction

Pancreatic cancer is difficult to target because its fibrotic microenvironment not only acts as a barrier for delivery of tumor cell targeting drugs, but it also generates an anti-inflammatory environment and prevents immunotherapy (*Balachandran et al., 2019*). One of the current paradigms for treatment of PDA focuses on combining chemotherapy with immune modulators that reprogram tumor-promoting macrophages toward a pro-inflammatory phenotype (*Bastea et al., 2019*; *Mitchem et al., 2013*; *Pandey and Storz, 2019*). A deeper understanding of the mechanisms that play a role in macrophage polarization can provide insights to develop such new interventions.

Genetic mouse models have shown that pancreatic ductal adenocarcinoma (PDA) most likely originates from precancerous pancreatic intraepithelial neoplasm (PanIN) lesions (reviewed in *Storz, 2017*). The development and progression of these early lesions is dependent on crosstalk between a multitude of host cells in their microenvironment. Of these, inflammatory (M1-polarized) and alternatively activated (M2-polarized) macrophages are the most consequential cell types.

The initial influx of macrophages, which induces local inflammation, occurs in response to an aberrant release of chemokines from pancreatic cells undergoing transformation (*Liou et al., 2015*). However, local inflammation alone is not an efficient driver of oncogenic progression and requires additional inflammatory signaling, genetic alterations, and downregulation of factors that maintain acinar cell identity (*Carrière et al., 2011*; *Cobo et al., 2018*; *Guerra et al., 2011*; *Guerra et al., 2007*).

Inflammatory macrophages contribute to pre-neoplastic lesion formation via secretion of inflammatory mediators, which regulate reorganization of the acinar microenvironment and initiate acinar-to-ductal metaplasia (ADM) (*Liou et al., 2015*; *Liou et al., 2013*; *Sawey et al., 2007*). While pro-inflammatory M1 macrophages are important for the initiation of precancerous lesions, this population dwindles and M2 macrophages become more predominant (*Liou et al., 2017*). These M2 macrophages are chitinase-like protein 3 (Ym1/*Chil3*), arginase-1 (Arg1), resistin-like alpha (Fizz1/*Retnla*), and interleukin-1 receptor antagonist protein (IL-1ra) positive and promote lesion growth, drive fibrogenesis, and block T-cell infiltration (*Bastea et al., 2019*; *Liou et al., 2017*). Later, at the tumor stage, alternatively activated macrophages represent approximately 85% of tumor-associated macrophages (TAMs) in the microenvironment (*Partecke et al., 2013*). For full-blown pancreatic cancer, tissue-resident macrophages have been suggested to shape fibrotic responses (*Zhu et al., 2017*), while infiltrating monocytes generate an immunosuppressive environment (*Zhang et al., 2017*; *Zhu et al., 2017*).

C-X-C motif chemokine 10 (CXCL10), also known as IFNγ-induced protein 10 (IP-10), acts through its cognate receptor C-X-C motif chemokine receptor 3 (CXCR3) (*Groom and Luster, 2011*) and regulates the chemotaxis of CXCR3$^+$ immune cells such as macrophages, T cells, and natural killer (NK) cells (*Luster and Ravetch, 1987*; *Tomita et al., 2016*; *Zhou et al., 2010*). With respect to cancer aggressiveness and patient prognosis, the presence of CXCL10 and CXCR3 has shown conflicting results depending on the type and stage of the disease (*Fulton, 2009*; *Jacquelot et al., 2018*; *Li et al., 2015*). In pancreatic cancer, both CXCL10 and CXCR3 are expressed in tumor tissue (*Delitto et al., 2015*), and their presence has been correlated with poor prognosis (*Liu et al., 2011*; *Lunardi et al., 2014*). However, the role of CXCL10/CXCR3 signaling during early development of the disease has not been addressed.

In our present study, we show that CXCL10, produced by precancerous lesions cells, is involved in the onset of inflammation by chemoattracting macrophages. We further show that CXCL10 signaling to CXCR3 is a key event for inflammatory macrophage identity and that inhibition of CXCL10/CXCR3 signaling leads to a polarization shift to an alternatively activated phenotype. In vivo, we demonstrate the importance of CXCL10/CXCR3 signaling in the maintenance of an inflammatory microenvironment, and that its blockage drives tumor progression.

## Results

### Pre-neoplastic ADM and PanIN lesions produce CXCL10

To identify factors that are released by precancerous lesion cells, we performed a cytokine/chemokine assay. Therefore, we used SM3 cells, which have been isolated from the precancerous epithelium of a KC mouse and form lesions with PanIN features when cultivated on extracellular matrix (*Agbunag et al., 2006*; *Liou et al., 2017*). In this screen limited to these in vitro lesion cells (*Figure 1—figure supplement 1A*), besides known factors such as C-C motif chemokine 5 (CCL5) and metalloproteinase inhibitor 1 (TIMP-1), we found strong expression of CXCL10, which has previously been identified as a chemoattractant for macrophages (*Tomita et al., 2016*; *Zhou et al., 2010*). We then used fluorescent in situ hybridization (FISH) to determine whether *Cxcl10* is produced in pancreatic precancerous lesion areas of *Ptf1a/p48$^{cre}$;LSL-Kras$^{G12D}$* (KC) mice. While *Cxcl10* was undetectable in normal adjacent acini (*Figure 1—figure supplement 1B*), we found significant expression of *Cxcl10* in ADM and PanIN1 lesions (*Figure 1A*). Quantification analyses of samples stained for *Cxcl10* mRNA by ISH indicated approximately fivefold higher expression in ADM than in PanIN (*Figure 1B*, *Figure 1—figure supplement 1C*). Next, we isolated primary pancreatic acinar cells from LSL-Kras$^{G12D}$ mice and adenovirally infected them with either GFP (control) or Cre-GFP, to test whether *Cxcl10* expression is upregulated during the KRas$^{G12D}$-driven ADM process (*Figure 1—figure supplement 1D*). However, expression of KRas$^{G12D}$ was unable to increase *Cxcl10* expression, indicating an external stimulus as a driver.

CXCL10 (also IP-10, interferon gamma-inducible protein 10) expression has previously been shown to be induced by interferon gamma (IFNγ) via activation of signal transducer and activator of transcription 1 (STAT1) (*Han et al., 2010*; *Luster and Ravetch, 1987*). Therefore, we tested if this pathway is active in PanIN cells. Treatment of SM3 cells with IFNγ induced an over 60-fold increase in *Cxcl10* mRNA (*Figure 1C*), as well as increased CXCL10 protein production (*Figure 1D*) and

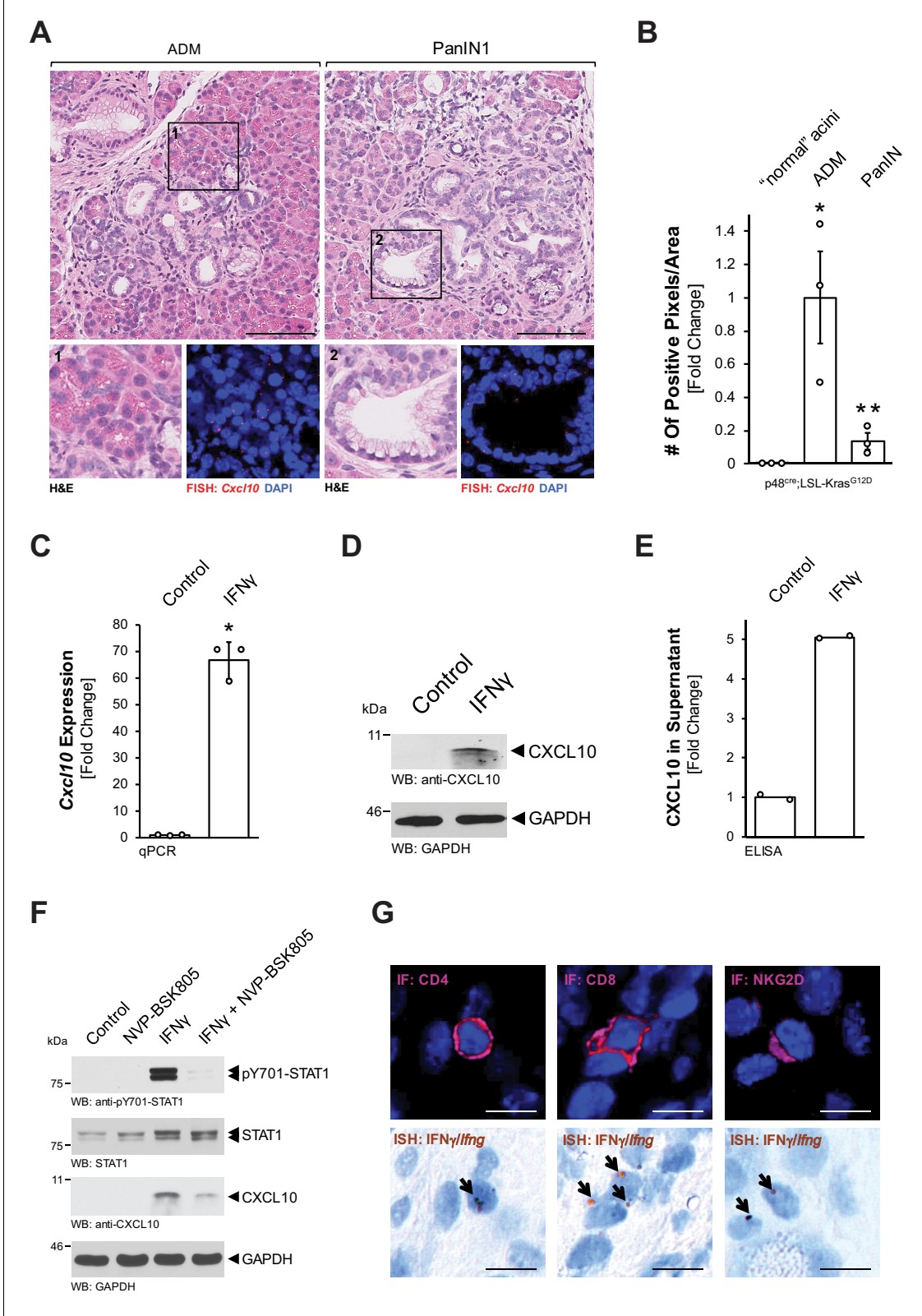

**Figure 1.** Pre-neoplastic ADM and PanIN lesions express CXCL10. (**A**) 'Normal' acinar cells (data shown in *Figure 1—figure supplement 1B*), ADM areas and PanIN lesions of pancreata from KC mice were analyzed for *Cxcl10* expression. Shown are representative pictures of H and E staining (overview and marked region) and FISH for *Cxcl10* mRNA expression (red dots) combined with DAPI in the marked regions. Images shown represent whole-slide analysis of staining. The scale bar represents 100 μm. (**B**) Quantification of relative *Cxcl10* expression (as determined by ISH shown in

*Figure 1 continued on next page*

Figure 1 continued

*Figure 1—figure supplement 1C*) in 'normal' acini, ADM, and PanIN regions of KC mice (n = 3 biological replicates) using a positive pixel algorithm on whole slides for each mouse analyzed, using the image scope software. Statistical analysis was done using the Student's t-test. *Statistical significance as compared to 'normal' acini (for ADM p-value = 0.023; PanIN p-value = ns), **Statistical significance for PanIN as compared to ADM (p-value=0.033). Error bars indicate standard deviation. (**C, D, E**) SM3 cells were stimulated with 10 ng/ml IFNγ for 4 days and an increase in CXCL10 expression was determined by qPCR (**C**), western blot (**D**), and in the media supernatants (**E**). For (**C, D**), results are representative of data from three independent, reproducible experiments. Statistical analysis was done using the Student's t-test. The asterisk indicates statistical significance (**C**: p-value < 0.0001; **E**: p-value = 0.0004). Error bars indicate standard deviation. (**E**) shows two biological repeats. (**F**) SM3 cells were treated with NVP-BSK805 (10 μM, 1 hr) and then stimulated with 10 ng/ml IFNγ for 24 hr. Samples were subjected to SDS-PAGE and analyzed by western blotting for pY701-STAT1, STAT1, and CXCL10 expression as indicated. Immunoblotting for GAPDH served as a control for equal loading. Results shown represent reproducible data obtained from three independent experiments. (**G**) Pancreata of KC mice were subjected to IF-IHC for CD4, CD8, and NKG2D combined with FISH for *Ifng*. Images shown are representative of IF and FISH done on 2 KC mice (biological replicates). The scale bar represents 10 μm.

The online version of this article includes the following source data and figure supplement(s) for figure 1:

**Source data 1.** Quantification of *Cxcl10* in KC tissue and IFNγ-stimulated SM3 cells (panels B, C, and E).
**Figure supplement 1.** CXCL10 expression in pancreatic precancerous lesion cells.
**Figure supplement 1—source data 1.** CXCL10 expression in Adeno-null-GFP and Adeno-cre-GFP infected cells (panel D).

---

secretion (*Figure 1E*). To test whether CXCL10 expression is indeed mediated through STAT1 signaling, we combined IFNγ stimulation with the pan-JAK inhibitor NVP-BSK805. We found that IFNγ stimulation led to phosphorylation of STAT1 at Y701 (activating phosphorylation), increased expression of CXCL10, and that pre-treatment with NVP-BSK805 inhibited IFNγ-induced pY701-STAT1 and CXCL10 expression (*Figure 1F*). T cells and NK cells are known IFNγ producers in the pancreatic microenvironment (*Brauner et al., 2010*; *Chapoval et al., 2001*; *Loos et al., 2009*). To determine whether these cells could be an in vivo source for IFNγ in our mouse model, we performed an ISH for *Ifng* combined with IHC for T-cell surface glycoprotein CD4 (CD4), T-cell surface glycoprotein CD8 (CD8), or NKG2-D type II integral membrane protein (NKG2D) markers. As expected from published data, we found both T cells and NK cells as a potential source for IFNγ (*Figure 1G*).

## Inflammatory macrophages are the recipients for CXCL10

With respect to early events leading to development of PDA, the influx of macrophages into the pancreas has been demonstrated following injury and during development and progression of pancreatic lesions (*Gea-Sorlí and Closa, 2009*; *Liou et al., 2015*). Moreover, CXCL10 has been demonstrated as a chemoattractant for macrophages along with other immune cells (*Liu et al., 2011*). This prompted us to test whether macrophages are responsive to CXCL10. We found that non-polarized peritoneal macrophages express high levels of the CXCL10 receptor *Cxcr3*, while M1-polarized (inflammatory) macrophages express moderate levels, and M2-polarized (alternatively activated) macrophages do not express this receptor (*Figure 2A*). Transwell invasion assays using both peritoneal and bone marrow-derived macrophages suggest that CXCL10 can act as a chemoattractant for M1-polarized macrophages (*Figure 2B*, *Figure 2—figure supplement 1A*). However, since tissue-resident macrophages have been attributed important roles in established pancreatic cancer (*Zhu et al., 2017*), we also determined if this population can be the recipients for CXCL10. Approximately 80% of tissue resident macrophages in normal mouse pancreas express CXCR3 (*Figure 2—figure supplement 1A*), but when isolated, these cells do not proliferate in response to CXCL10 (*Figure 2—figure supplement 1B*). In sum, our in vitro data suggests that CXCL10 may drive the chemoattraction of inflammatory macrophages to the pancreas.

Next, we determined if pancreatic macrophages or T cells express CXCR3 in KC mice. Therefore, we sorted for pancreatic CD3+ or F4/80+ cells and then for the presence of CXCR3. We found that approximately 40% of pancreatic macrophages in KC mice express CXCR3 (*Figure 2C*). Moreover, an in situ IF-IHC analysis of pancreata of KC mice indicated that inflammatory (F4/80+;pY701-STAT1+) macrophages express CXCR3, while alternatively activated (F4/80+;Ym1+) macrophages do not express this receptor (*Figure 2D*), which also confirmed above in vitro data. An overlay between an ISH for *Cxcr3* and IF-IHC for inflammatory macrophages (CD68+;iNOS+) in human patient tumors showed that ~70% of CXCR3+ cells are M1 macrophages and confirmed this population as a potential recipient for CXCL10 (*Figure 2E,F*).

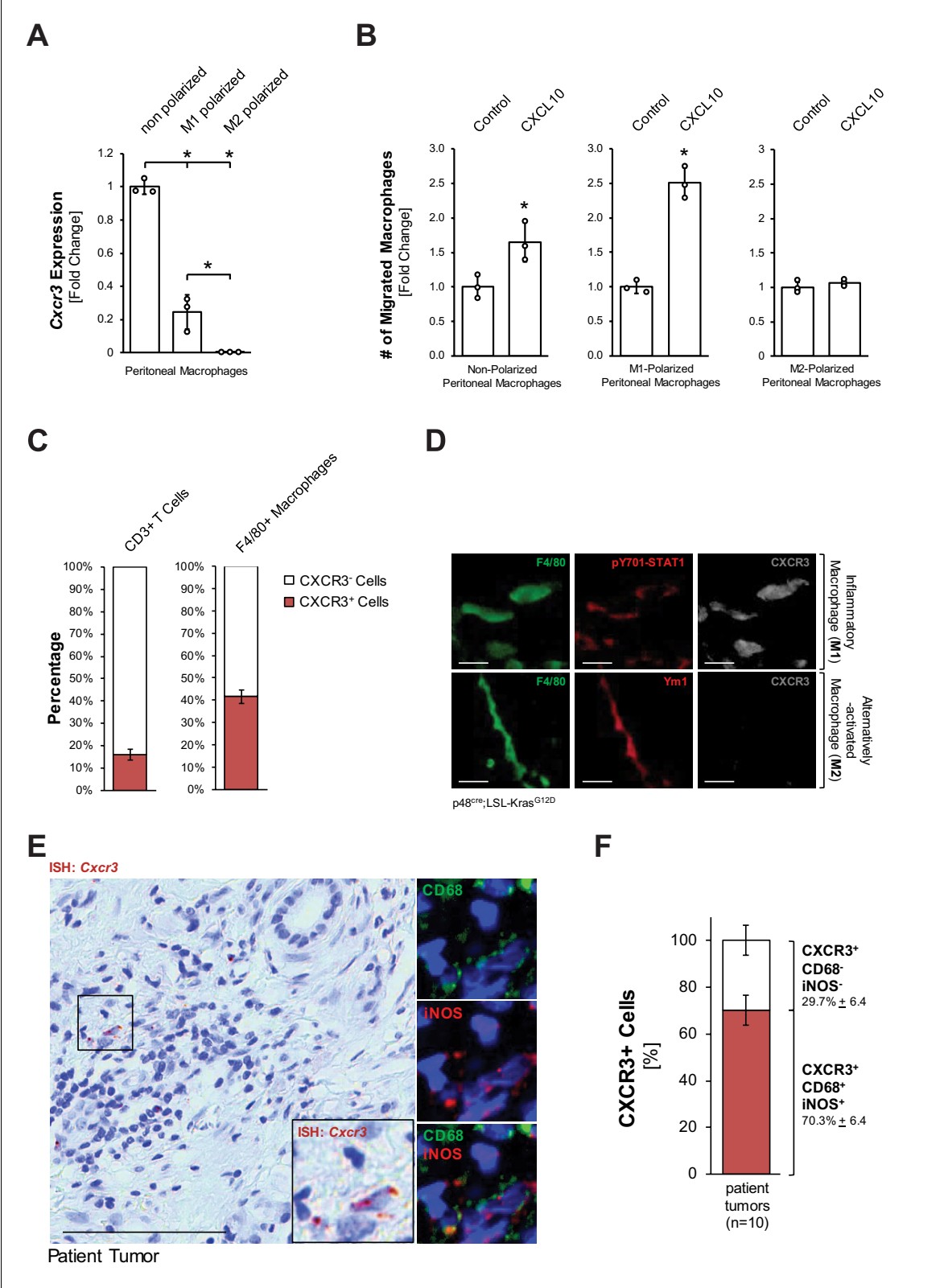

**Figure 2.** Inflammatory macrophages are the recipients for CXCL10. (**A**) Primary peritoneal macrophages were isolated and either left non-polarized or were polarized to M1 and M2 macrophages. CXCR3 expression was determined by qPCR. Results shown are representative of reproducible data from three independent experiments. Statistical analysis using the Student's t-test indicates significance (marked by an asterisk). Error bars indicate standard deviation. (**B**) Transwell assay. $0.5 \times 10^5$ non-polarized, M1- or M2-polarized peritoneal macrophages were plated into transwell inserts. 500 ng/ml

*Figure 2 continued on next page*

*Figure 2 continued*

CXCL10 in media was placed in the bottom wells, and chemoattraction of macrophages was determined after 20 hr. Data shown here represents reproducible results from peritoneal macrophages obtained from three different mice (biological replicates). Statistical analysis using the Student's t-test indicates significance (marked by an asterisk) for nonpolarized (p-value=0.034) and M1 (p-value=0.0003) macrophages. Error bars indicate standard deviation. (C) CD3$^+$ T cells and F4/80$^+$ macrophages were sorted from digested pancreas of KC mice using FACS and analyzed for the expression of CXCR3. The bars indicate the percentage of CXCR3$^+$ cells among the two cell types sorted. Data represents analyses done on three mice (biological replicates). Statistical analysis was done using Student's t-test. Error bars indicate standard deviation. (D) CXCR3 expression in M1 (F4/80$^+$; pSTAT1$^+$) or M2 (F4/80$^+$;Ym1$^+$) macrophages in pancreatic tissue of KC mice was determined by immunofluorescence. Images shown here represent whole slide analysis of staining done on the tissue of 2 KC mice (biological replicates). The scale bar indicates 10 µm. (E) Patient tumor tissue stained by ISH for *Cxcr3* and overlaying immunofluorescence for inflammatory (CD68$^+$,iNOS$^+$) macrophages. Images shown represent whole slide analysis of staining done on the tissue from 10 patient samples. The scale bar indicates 100 µm. (F) Quantification of *Cxcr3*$^+$;CD68$^+$;iNOS$^+$ cells in patient samples (n = 10). Statistical analysis was done using Student's t-test. Error bars indicate standard deviation.

The online version of this article includes the following source data and figure supplement(s) for figure 2:

**Source data 1.** *Cxcr3* expression and migration in response to CXCL10 for polarized macrophages, CXCR3 expression in T cells and macrophages, and quantification of CXCR3+ M1 macrophages (panels A, B, C, and F).

**Figure supplement 1.** M1-polarized BMD macrophages are chemoattracted to CXCL10, and tissue resident macrophages express CXCR3 but do not proliferate in response to CXCL10.

**Figure supplement 1—source data 1.** Quantification of macrophage migration and proliferation in response to CXCL10, and percentage of macrophages which are CXCR3+ (panels A-C).

## CXCL10/CXCR3 signaling maintains the inflammatory phenotype of macrophages

Next, we tested the impact of blocking CXCL10/CXCR3 signaling on the inflammatory macrophage population. We isolated peritoneal primary macrophages, polarized them to inflammatory (F4/80$^+$; iNOS$^+$) macrophages (*Figure 3—figure supplement 1A*), and then blocked CXCL10/CXCR3 signaling with a CXCR3 neutralizing antibody (CXCR3 NAB). We found that upon CXCR3 neutralization, M1 macrophages, although still iNOS positive, start to express Ym1 (*Figure 3A*), which we previously have identified as a *bona fide* marker for M2 alternatively activated macrophages in precancerous abnormal pancreas lesions (*Bastea et al., 2019*). A more thorough analysis using qRT-PCR indicated a significant decrease in CD38, which is a distinct M1 macrophage marker (*Jablonski et al., 2015*), and an upregulation of Ym1, Arg1 and Fizz1 (*Figure 3B*), which are all markers of the alternatively activated macrophage population that previously has been shown to drive fibrosis during pancreatic lesion progression (*Liou et al., 2017*). CXCR3 neutralization in M1 macrophages also slightly increased *Il4ra* and *Stat6* mRNA expression, and increased nuclear localization of active Y641-phosphorylated STAT6 (*Figure 3—figure supplement 1B,C*), which are additional indicators of M2 polarization (*Yushi et al., 2016*).

The interferon regulatory factors (IRFs) IRF4 and IRF5 have been shown to be key regulators of macrophage polarization (*Bastea et al., 2019*; *Günthner and Anders, 2013*). M1-polarized macrophages subjected to CXCR3 inhibition upregulated IRF4, a transcription factor for M2 macrophage polarization, and downregulated the M1 transcription factor IRF5 (*Figure 3—figure supplement 1D*), indicating a potential mechanism of how pancreatic macrophage populations may shift when CXCR3 is blocked.

To test whether neutralization of CXCL10/CXCR3 signaling also decreases inflammatory macrophages in vivo, we treated KC mice with a CXCR3 neutralization antibody (CXCR3 NAB) or an isotype control IgG antibody over a period of 9 weeks (*Figure 3—figure supplement 1E*). While overall numbers of macrophages remained unchanged (*Figure 3—figure supplement 1F*), IF-IHC analyses for F4/80$^+$;pY701-STAT1$^+$ cells indicated a >50% decrease in inflammatory macrophages in the CXCR3 NAB-treated mice (*Figure 3C,D*).

## Neutralization of CXCL10/CXCR3 signaling increases alternatively activated pancreatic macrophages, SMA-positive fibroblasts, and areas of pancreatic lesions

We then determined if the decrease in inflammatory macrophages after neutralization of CXCL10/CXCR3 signaling can lead to an increase in Ym1$^+$ alternatively activated macrophages. The presence of Ym1$^+$ macrophages in the pancreas was increased by approximately threefold when KC mice

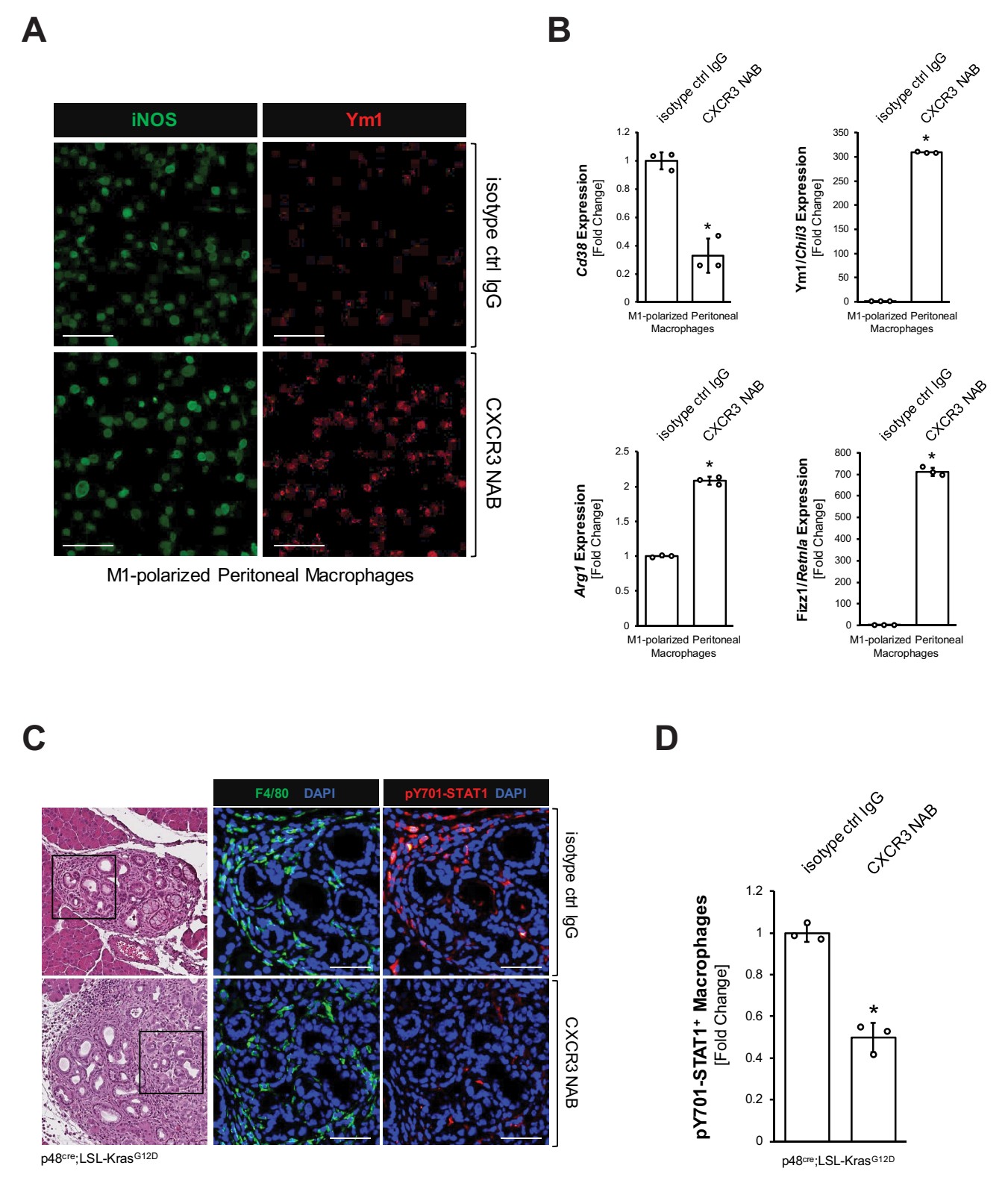

**Figure 3.** CXCL10/CXCR3 signaling maintains the inflammatory phenotype of macrophages. (**A**) Peritoneal macrophages were isolated, polarized to M1, and then treated with 500 µg/ml CXCR3 NAB or isotype control IgG. After 48 hr treatment, samples were analyzed for expression of iNOS or Ym1 using immunofluorescence. Images shown are representative of three independent experiments done on peritoneal macrophages obtained from three mice (biological replicates). The scale bar indicates 100 µm. (**B**) Peritoneal macrophages were isolated, polarized to M1, and treated with 500 µg/ml

*Figure 3 continued on next page*

*Figure 3 continued*

CXCR3 NAB or isotype control IgG. After 48 hr, samples were analyzed by qPCR for expression of M1 macrophage marker *Cd38* and M2 macrophage markers Ym1/*Chil3*, *Arg1*, and Fizz1/*Retnla*. Results shown are representative of three independent experiments done on peritoneal macrophages obtained from three mice (biological replicates). Statistical analysis using the Student's t-test indicates significance (marked by an asterisk, *Cd38*: p-value=0.0009, Ym1/*Chil3*: p-value<0.0001, *Arg1*: p-value<0.0001, Fizz1/*Retnla*: p-value<0.0001). Error bars indicate standard deviation. (C) Pancreatic abnormal areas from KC mice treated with CXCR3 NAB or isotype control IgG were analyzed for presence of inflammatory macrophages (co-immunofluorescence for F4/80 and pY701-STAT1). Shown is a representative area from staining and analysis done on three mice per group. The H and E staining highlights the region analyzed. The scale bar indicates 50 μm. (D) Quantification of pY701-STAT1$^+$ macrophages in pancreata from KC mice (n = 3 mice per treatment group) treated with CXCR3 NAB or isotype control IgG. Cells were counted in three representative fields per mouse. The *arcsin* transformation was done on the proportion of macrophages which were pY701-STAT1$^+$. Statistical analysis using the Student's t-test indicates significance between biological replicates (indicated by an asterisk, p-value=0.0004). Error bars indicate standard deviation.

The online version of this article includes the following source data and figure supplement(s) for figure 3:

**Source data 1.** qPCR for *Cd38*, Ym1/*Chil3*, *Arg1*, and Fizz1/*Retnla* in M1 polarized macrophages treated with CXCR3 NAB or isotype control IgG, and quantification of pY701-STAT1+ macrophages in KC mice treated with CXCR3 NAB or isotype control IgG (panels B and D).

**Figure supplement 1.** Neutralization of CXCR3 shifts macrophages to M2 polarization.

**Figure supplement 1—source data 1.** qPCR for *Il4ra*, *Stat6*, *Irf4*, and *Irf6* in M1 polarized peritoneal macrophages treated with CXCR3 NAB or isotype control IgG, and quantification of macrophages in KC mice treated with CXCR3 NAB or isotype control IgG (panels B, D, and F).

---

were treated with a CXCR3 NAB, as compared to treatment with the isotype control IgG (*Figure 4A,B*). Similar effects on this macrophage population were observed when KC mice were treated with a CXCL10 NAB, as compared to treatment with the isotype control IgG (*Figure 4—figure supplement 1A–C*).

Previously, Ym1$^+$ macrophages in the KC animal model have been shown to drive fibrosis during pancreatic cancer development (*Liou et al., 2017*), and after neutralization of CXCR3, or alternatively neutralization of CXCL10, we observed a correlating increase in fibrosis (*Figure 4C,D* and *Figure 4—figure supplement 1D,E*). In line with increased presence of alternatively activated macrophages and increased fibrosis, abnormal areas (lesions and stroma) increased approximately twofold when KC mice were treated with CXCR3 NAB (*Figure 4E,F*). Of note, neutralization of CXCR3 in non-transgenic (ntg) mice did not show any change in the normal pancreatic tissue (*Figure 4E*). Quantification of different lesion types from both treatment groups indicated slight, but not statistically significant differences in the distribution of ADM (80.7 ± 3.8% in control-treated *versus* 75.2 ± 4% CXCR3 NAB-treated) and PanIN1 (19.3 ± 3.8% in control-treated *versus* 24.6 ± 4% CXCR3 NAB-treated) lesions (*Figure 4G*).

## Neutralization of CXCL10/CXCR3 signaling does not significantly affect the presence of T cells

In addition to macrophages, T cells can be chemoattracted via CXCR3 signaling (*Dufour et al., 2002*). Since approximately 15% of CD3$^+$ T cells express CXCR3 in our KC animal model (see *Figure 2C*), we also tested the effect of CXCL10/CXCR3 neutralization on T-cell populations in pancreata of KC mice. After neutralization of CXCR3, we detected a slight (approximately 20%) decrease in pancreatic CD3$^+$ cells (*Figure 4—figure supplement 2A*), but did not observe any changes in NK (NKG2D$^+$;CD3$^-$) cells (*Figure 4—figure supplement 2B*) or production of IFNγ (*Figure 4—figure supplement 2C*). Moreover, neutralization of CXCL10 did not result in a decrease of CD3$^+$;CD4$^+$ or CD3$^+$;CD8$^+$ T cells (*Figure 4—figure supplement 2D,E*).

## T cells do not contribute to CXCL10/CXCR3-mediated effects on macrophage populations or abnormal pancreatic lesions

To test whether effects on macrophage populations or abnormal pancreatic lesion areas are due to the presence of T cells, we performed a T-cell depletion followed by CXCR3 neutralization (*Figure 4—figure supplement 3A*). T cells in KC mice were depleted using previously described antibodies that target CD8α and CD4 (*Laky and Kruisbeek, 2016*). Regardless of T cells being depleted or present (*Figure 4—figure supplement 3B*), CXCR3 neutralization increased abnormal pancreatic area (*Figure 4—figure supplement 3C*), decreased M1 macrophages (*Figure 4—figure supplement 3D*), and increased Ym1$^+$ macrophages (*Figure 4—figure supplement 3E*). Moreover,

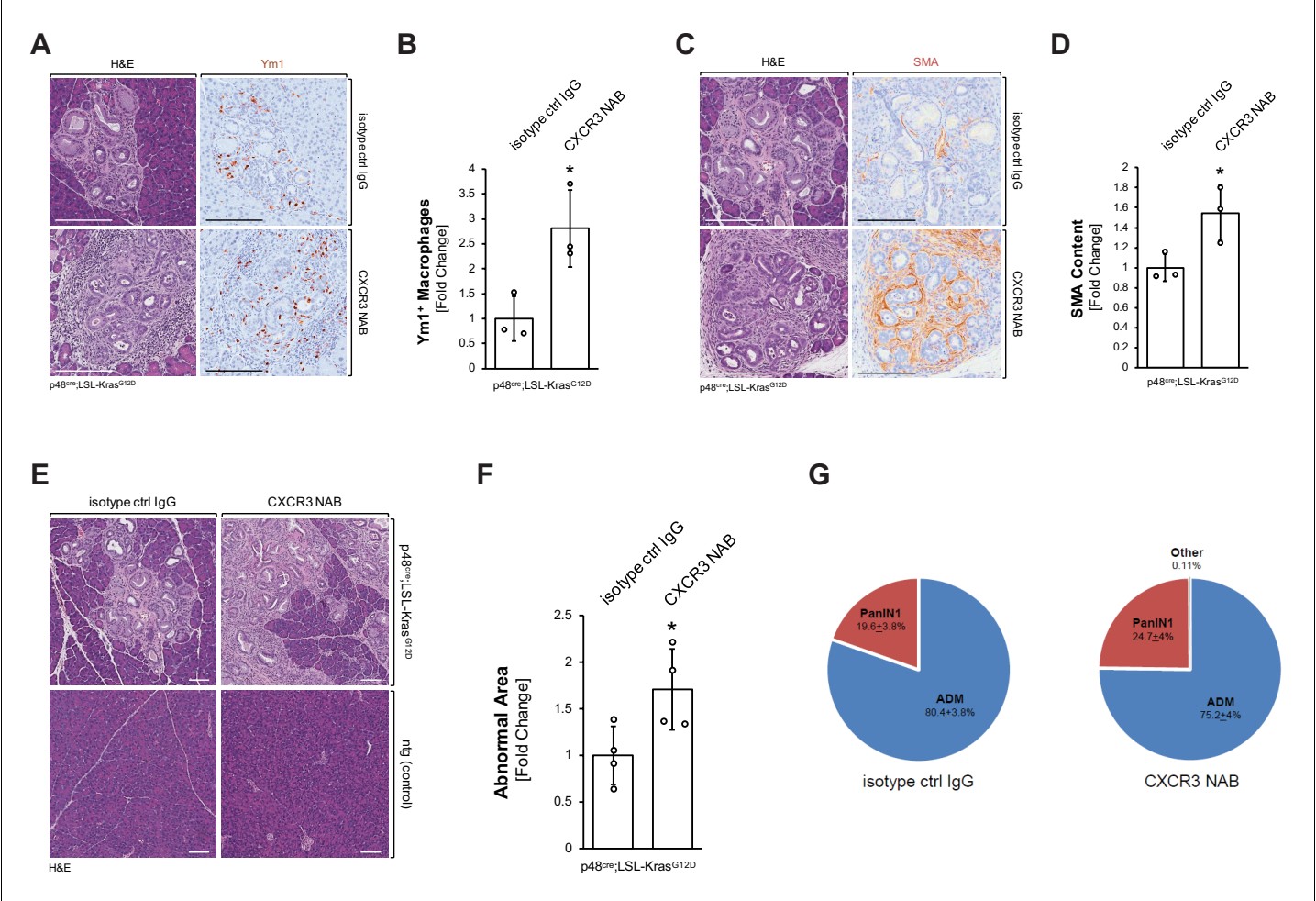

**Figure 4.** Neutralization of CXCR3 signaling increases alternatively activated pancreatic macrophages and progresses pancreatic lesions. (**A**) Pancreatic abnormal areas from KC mice treated with CXCR3 NAB or isotype control IgG were analyzed by IHC for presence of Ym1+ macrophages. Shown is a representative area and H and E staining on serial sections of the tissue. The scale bar indicates 100 μm. (**B**) Quantification of Ym1+ macrophages in pancreata from KC mice (n = 3 mice per treatment group) treated with CXCR3 NAB or isotype control IgG. Ym1+ cells were counted in three representative fields per mouse, and reported as the fold change relative to the average of the isotype control IgG treatment group. Statistical analysis was done using the Student's t-test. The asterisk indicates statistical significance (p-value=0.025). Error bars indicate standard deviation. (**C**) Pancreatic abnormal areas from KC mice treated with CXCR3 NAB or isotype control IgG were analyzed by IHC for the fibrosis marker smooth muscle actin (SMA). Shown is a representative area and H and E staining on serial sections of the tissue. The scale bar indicates 100 μm. (**D**) Quantification of SMA content in pancreata from KC mice (n = 3 per treatment group) treated with CXCR3 NAB or isotype control IgG. Quantification was performed with the Image scope positive pixel algorithm. Abnormal areas were manually traced on the tissue, and the algorithm was run. The resulting pixel values were divided by the area analyzed to obtain staining per area of abnormal tissue. Statistical analysis was done using the Student's t-test. The asterisk indicates statistical significance (p-value=0.038). Error bars indicate standard deviation. (**E**) Representative images of H and E stained pancreata from non-transgenic (ntg) or KC mice treated with CXCR3 NAB or isotype control IgG. The scale bar indicates 100 μm. (**F**) Quantification of the abnormal pancreatic surface area in pancreata from KC mice (n = 4 per treatment group) treated with CXCR3 NAB or isotype control IgG. For quantification, abnormal areas were manually traced out and values were normalized to the total pancreatic area to obtain abnormal area per area of pancreatic tissue analyzed. For statistical analysis, data were transformed via *arcsin* transformation before a *t*-test was performed. The asterisk indicates statistical significance (p-value=0.037). Error bars indicate standard deviation. (**G**) Pie graph showing the percentage distribution of the types of lesions found in the abnormal surface areas of mice from both treatment groups. Percentages (with standard error) shown are from analysis done on four mice per group.

The online version of this article includes the following source data and figure supplement(s) for figure 4:

**Source data 1.** Quantification of Ym1, SMA, abnormal area, and lesion type in CXCR3 NAB and isotype control IgG treated KC mice (panels B, D, F, and G).

**Figure supplement 1.** Neutralization of CXCL10 increases alternatively activated macrophages and fibrosis.

**Figure supplement 1—source data 1.** Quantification of Ym1+ cells and SMA content in KC mice treated with CXCL10 NAB or isotype control IgG (panels C and E).

*Figure 4 continued on next page*

*Figure 4 continued*

**Figure supplement 2.** Neutralization of CXCL10/CXCR3 signaling does not significantly affect the presence of T cells.

**Figure supplement 2—source data 1.** Quantification of T cells, NK cells, and IFNγ/*Ifng* in KC mice treated with CXCL10 NAB or isotype control IgG (panels A-C), and T cell quantification in KC mice treated with CXCL10 NAB or isotype control IgG.

**Figure supplement 3.** T cells do not contribute to CXCL10/CXCR3-mediated effects on macrophage populations or abnormal pancreatic lesions.

**Figure supplement 3—source data 1.** Quantification of T cells, abnormal tissue area, macrophages, and IFNγ/*Ifng* in T cell depletion experiments (panels B-F), and secretion of CXCL10 in lentivirally-infected SM3 PanIN organoids (panel H).

T-cell depletion led to a slight but non-significant decrease in *Ifng*, suggesting that T cells are not the only source for IFNγ in this experimental system (*Figure 4—figure supplement 3F*).

To rigorously test a direct effect of CXCL10/CXCR3 signaling on macrophages in a system absent of T cells, we expressed CXCL10 in SM3 PanIN cells and implanted them into the pancreas of T cell-deficient athymic nude mice (*Figure 4—figure supplement 3G*). Therefore, we infected SM3 PanIN cells (described in *Liou et al., 2017*) with a lentivirus carrying CXCL10 (SM3-CXCL10) or a control lentivirus carrying eGFP (SM3-control), tested them for secretion of CXCL10 (*Figure 4—figure supplement 3H*), and generated PanIN organoids by plating them in Matrigel for 2 days (*Figure 4—figure supplement 3I*). Organoids were recovered from Matrigel and, together with activated stellate cells, implanted into the pancreas of nude mice. Two weeks after implantation, SM3-CXCL10 implanted mice had approximately half as much abnormal tissue area (lesions and fibrosis) as compared to SM3-control implanted mice (*Figure 5A,B*). Analysis of SMA content showed significantly reduced fibrosis in the SM3-CXCL10 implanted mice (*Figure 5C*). As expected from the reverse experiment in which we had blocked CXCR3 (*Figure 3D*), expression of CXCL10 increased the presence of F4/80⁺ macrophages (*Figure 5D*). Further analysis of macrophage polarization types indicated an increase of M1 and a decrease of M2 phenotypes when CXCL10 was present (*Figure 5E*).

## Discussion

While inflammatory and alternatively activated macrophages have important functions in different stages of pancreatic cancer development (*Stone and Beatty, 2019*; *Storz, 2017*), little is known about how these populations are regulated after injury or an oncogenic event, and how they cross-talk with lesion cells or other cells of the lesion microenvironment. In this study, we delineate the importance of CXCL10/CXCR3 signaling during the early phase of pancreatic cancer development.

The CXCL10/CXCR3 signaling axis has been implicated in generating inflammation in various diseases including pancreatitis (*Lee et al., 2017*; *Liu et al., 2011*; *Singh et al., 2007*). This is because it regulates the chemotaxis of CXCR3⁺ immune cells such as macrophages, T cells, and NK cells (*Luster and Ravetch, 1987*; *Taub et al., 1993*; *Tomita et al., 2016*; *Zhou et al., 2010*). With respect to cancer, gastric cancer patients with upregulated CXCR3 expression showed better survival (*Chen et al., 2019*; *Li et al., 2015*). Also, high expression of CXCL10 and other CXCR3 ligands in ovarian, esophageal, and non-small cell lung carcinoma indicate favorable prognoses (*Bronger et al., 2016*; *Cao et al., 2017*; *Sato et al., 2016*). In contrast, in pancreatic cancer, expression of both CXCL10 and CXCR3 in tumor tissue has been correlated with a poor prognosis (*Liu et al., 2011*; *Lunardi et al., 2014*) mostly due to increased chemoresistance (*Delitto et al., 2015*). However, the role of CXCL10/CXCR3 signaling during early development of the disease has not been addressed.

Using the KC mouse model, we show that CXCL10 is produced by cells of precancerous ADM and PanIN1 lesions (*Figure 1A,B*). CXCL10 is also known as interferon gamma-inducible protein-10 (IP-10) (*Luster et al., 1985*; *Ohmori and Hamilton, 1993*), suggesting IFNγ as a potential factor sufficient to drive CXCL10 expression. Indeed, SM3 PanIN cells showed upregulation of CXCL10 expression in response to IFNγ (*Figure 1C–E*). This was mediated through JAK-STAT1 signaling (*Figure 1F*), which has been previously described to be upregulated and activated via IFNγ in acinar cells (*Gallmeier et al., 2005*). Many cell types, including T cells and NK cells in the pancreatic microenvironment, have been shown to secrete IFNγ (*Brauner et al., 2010*; *Castro et al., 2018*; *Corthay et al., 2005*; *Fogar et al., 2011*), and analysis for CD4⁺, CD8⁺, and NKG2D⁺ cells indicated T cells and NK cells in the vicinity of lesions as a potential source for IFNγ in our model (*Figure 1G*).

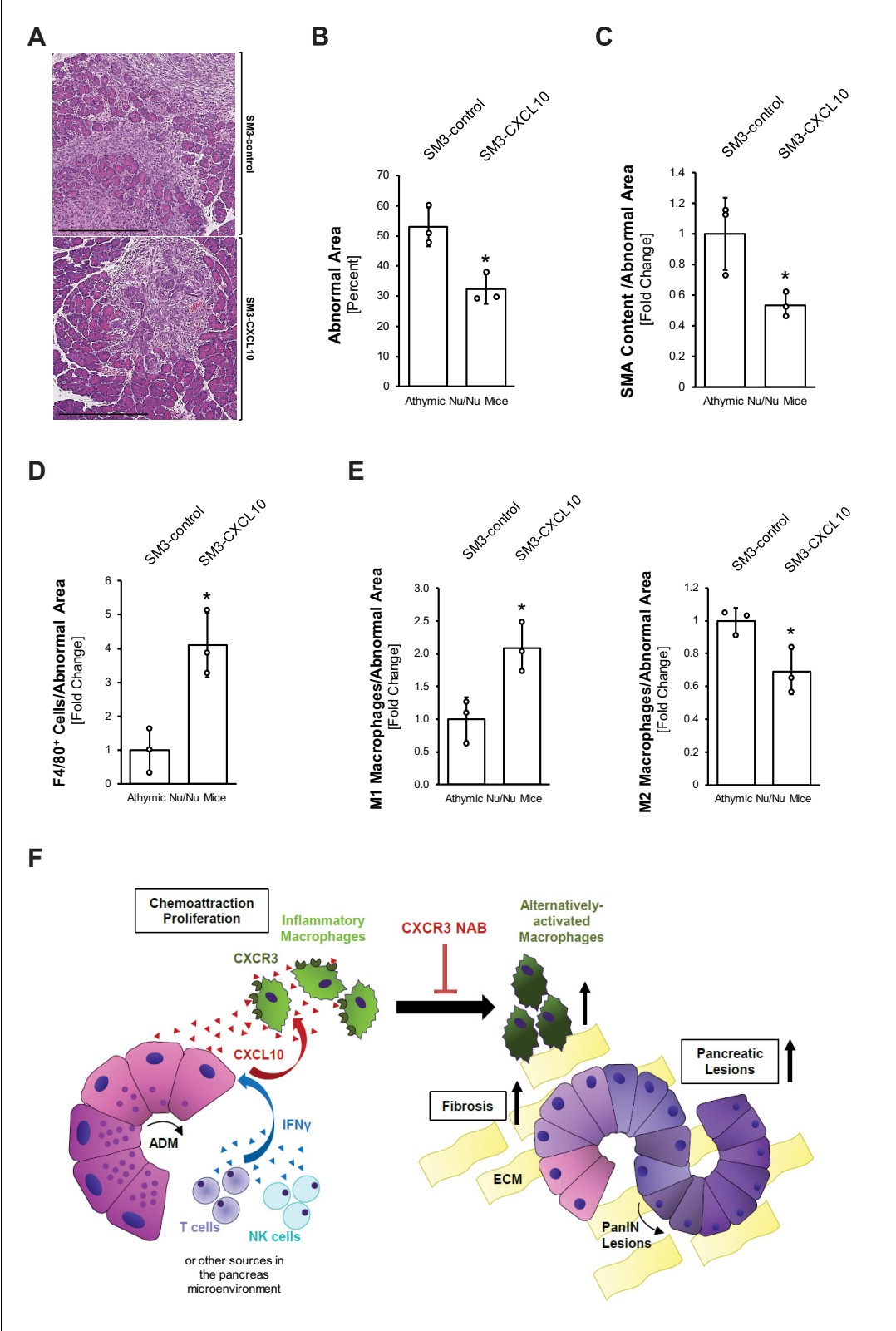

**Figure 5.** Overexpression of CXCL10 in pancreatic lesions increases inflammatory macrophages and decreases lesion formation in the pancreas. (**A**) Athymic nude mice were orthotopically implanted with PanIN organoids obtained from lentivirally infected SM3-CXCL10 or SM3-control cells (see *Figure 4—figure supplement 3G–I*). Mice were euthanized 2 weeks post-surgery, and pancreatic tissue was analyzed. Shown are representative images of abnormal areas in the pancreas. The scale bar represents 500 µm. (**B**) Abnormal areas (lesions and stroma) were manually traced for *Figure 5 continued on next page*

**Figure 5 continued**

quantification and normalized to total pancreatic area analyzed (n = 3 mice per group). For statistical analysis, data were transformed via *arcsin* transformation before a *t*-test was performed. The asterisk indicates statistical significance (p-value=0.011). Error bars indicate standard deviation. (C) Fibrotic content was analyzed using SMA as a marker (n = 3 mice per group). Quantification was performed with the Image scope positive pixel algorithm as described in *Figure 4* and normalized to the total areas analyzed. Statistical analysis was done using the Student's t-test. The asterisk indicates statistical significance (p-value=0.032). Error bars indicate standard deviation. (D) Tissue was analyzed for infiltration of total macrophages between groups using F4/80 as a marker (n = 3 mice per group). Staining was quantified using the positive pixel algorithm and normalized to the total areas analyzed. Statistical analysis was done using the Student's t-test. The asterisk indicates statistical significance (p-value=0.0096). Error bars indicate standard deviation. (E) Detailed analysis of tissue with F4/80 and pY701-STAT1 (M1 macrophage population) and Ym1 (M2 macrophage population). Quantification was performed either manually (M1) on three representative fields for each mouse tissue analyzed or using the positive pixel algorithm (M2) and then normalized to the total areas analyzed (n = 3 mice per group). Statistical analysis was done using Student's t-test. The asterisk indicates statistical significance (M1: p-value=0.019, M2: p-value=0.028). Error bars indicate standard deviation. (F) Schematic diagram of how CXCL10/CXCR3 signaling impacts pancreatic lesion progression. IFNγ released from immune cells (T and NK) in the pancreatic tissue stimulates CXCL10 release from early lesions (ADM, PanIN1). CXCL10 stimulates chemoattraction and proliferation of peritoneal macrophages and helps maintain their inflammatory phenotype. Blocking the ligand–receptor interaction with a CXCR3 NAB leads to loss of M1 identity, resulting in an increase in the Ym1$^+$ macrophage population, along with more lesions and a higher fibrotic content.

The online version of this article includes the following source data for figure 5:

**Source data 1.** Quantification of abnormal area, SMA, F4/80, M1 macrophages, and M2 macrophages in athymic nude mice orthotpically implanted with lentivirally infected SM3-CXCL10 or SM3-control PanIN organoids (panels B-E).

Our in vitro data implicate peritoneal or bone-marrow-derived inflammatory macrophages as major recipients for CXCL10, where CXCL10 can act as a chemoattractant (*Figure 2B*, *Figure 2—figure supplement 1A*). This is important because influx of macrophages is a major driver of pancreatic inflammation (*Gea-Sorlí and Closa, 2009*). This correlates with in vivo data showing that in the KC model CXCR3 is expressed by pancreatic inflammatory macrophages, but not by Ym1$^+$ M2 macrophages (*Figure 2D*). In patient tissue, we found that approximately 70% of CXCR3 expressing cells are inflammatory (CD68$^+$;iNOS$^+$) macrophages (*Figure 2E,F*).

Unlike other factors contributing to macrophage chemoattraction into the pancreas, such as sICAM1 (*Liou et al., 2015*), the expression of CXCL10 from acinar cells is not driven by the activation of oncogenic KRas (*Figure 1—figure supplement 1D*). This finding is supported by other studies that have shown CXCL10 upregulation in chronic pancreatitis patients, which occurs in the absence of oncogenic mutations (*Singh et al., 2007*).

The importance of CXCL10/CXCR3 signaling in inflammatory macrophages is underscored by the fact that blocking this interaction causes them to lose their identity, as evident by upregulation of markers for alternatively activated macrophages (*Figure 3A–C*). Even though neutralizing CXCR3 did not simultaneously decrease iNOS expression while increasing M2 markers (*Figure 3A*, *Figure 3—figure supplement 1B*), there was a marked drop in *Cd38*, a distinct marker for M1 polarized macrophages (*Jablonski et al., 2015*), indicating a loss in M1 identity (*Figure 3B*). Mechanistically, this shift in polarization upon CXCR3 neutralization may be explained by upregulation of the M2 transcription factor *Irf4* and downregulation of the M1 transcription factor *Irf5* (*Figure 3—figure supplement 1D*). Interestingly, the resulting alternatively activated macrophage population is characterized by expression of markers (Ym1, Fizz1, Arg1) that previously have been attributed to a subset of pancreatic macrophages (usually described as Ym1$^+$ macrophages) that drive fibrinogenesis and tumor progression (*Liou et al., 2017*).

In support of this explant data, neutralizing CXCR3 in vivo also led to a decrease in inflammatory (F4/80$^+$;pY701-STAT1$^+$) macrophages (*Figure 3C,D*) and an increase in Ym1$^+$ macrophages (*Figure 4A,B*). This accompanied increased fibrosis at the pancreatic lesion areas and more abnormal regions in the pancreas overall (*Figure 4C, D, and F*). Our data is in line with a recent study in which it was reported that *Cxcr3$^{−/−}$* mice show an abundance of M2 macrophages in breast tumors (*Oghumu et al., 2014*). Specifically, similar to what we observed after CXCR3 neutralization, *Cxcr3$^{−/−}$* macrophages from these mice had reduced ability to upregulate iNOS and showed a predisposition for M2 polarization (*Oghumu et al., 2014*). Since there are other ligands described that can engage the receptor CXCR3 (*Van Raemdonck et al., 2015*), we also neutralized CXCL10 in KC mice and found similar trends of increased presence of Ym1$^+$ macrophages and increased fibrosis (*Figure 4—figure supplement 1*). In a reverse experiment, enhanced expression of CXCL10 in PanIN

cells significantly decreased overall abnormal lesion areas, fibrosis, and presence of M2 macrophages, but increased the presence of M1 macrophages (*Figure 5*).

CXCL10 is known to chemoattract cell types other than macrophages, mainly T cells, and approximately 15% of pancreatic T cells in KC mice are positive for CXCR3 (*Figure 2C*). However, our in vivo study in which we overexpressed CXCL10 in PanIN cells and observed reverse effects to CXCR3 or CXCL10 neutralization was performed in nude mice which lack T cells (*Figure 4—figure supplement 2*). In addition, depletion of CD4$^+$ and CD8$^+$ T cells in immunocompetent KC mice had no impact on total abnormal tissue area or M1 and M2 macrophage abundance after CXCR3 neutralization (*Figure 4—figure supplement 3A–E*). Taken together these data suggest no significant role for T cells in CXCL10/CXCR3-driven development of pancreatic cancer.

During pancreatic cancer development, inflammatory macrophages are predominantly located at ADM lesions, driving the transdifferentiation process (*Liou et al., 2015*; *Liou et al., 2013*). Our data now indicate that cells undergoing ADM (and to some extent low-grade PanIN) express CXCL10 to chemoattract inflammatory macrophages and that CXCL10/CXCR3 signaling is crucial for the sustenance of their inflammatory identity (*Figure 5F*). More progressed PanIN1 lesions have been shown to express the anti-inflammatory cytokine interleukin-13 (IL-13), which causes a polarization shift from M1 toward an alternatively activated, Ym1$^+$ phenotype (*Liou et al., 2017*). Given the importance of CXCL10/CXCR3 interaction in the maintenance of the inflammatory identity of macrophages in our study, the reduced CXCL10 expression by PanIN as compared to ADM lesions (*Figure 1A,B*) may leave macrophages more susceptible to M2 polarization by cytokines like IL-13.

Considering our results, use of agonists for the receptor CXCR3 at stages of low-grade lesions may be useful to modulate macrophage polarization in the microenvironment such that a predominantly inflammatory population (M1) can be sustained. However, it needs to be noted that in pancreatic cancer expression of both CXCL10 and CXCR3 in tumor tissue have been correlated with metastasis and poor prognosis (*Cannon et al., 2020*; *Hirth et al., 2020*; *Liu et al., 2011*; *Lunardi et al., 2014*; *Romero et al., 2020*). Therefore, it is unclear if treatment of pancreatic tumors with a CXCR3 agonist will result in a polarization switch of tumor-associated macrophages that renders the lesion microenvironment less supportive for tumors, and increases efficiency of chemotherapy, or if it has a tumor-promoting effect. This will be addressed in future studies.

## Materials and methods

### Cells, antibodies, and reagents

SM3 primary duct-like cells were isolated from pancreata of 6 week old KC mice as previously described (*Agbunag et al., 2006*; *Liou et al., 2017*). The genotype of these cells has been verified, and cells are routinely tested for mycoplasma infection. SM3 cells were maintained in DMEM/F12 media (Sigma-Aldrich, St. Louis, MO) containing 5% Nu Serum IV culture supplement (Corning, Corning, NY), 25 µg/ml bovine pituitary extract (Gibco/Thermo Scientific, Waltham, MA), 20 ng/ml EGF, 0.1 mg/ml soybean trypsin inhibitor type I (AMRESCO, Solon, OH), 5 mg/ml D-glucose (Sigma-Aldrich), 1.22 mg/ml nicotinamide (Sigma-Aldrich), 5 nM triiodo-L-thyronine (Sigma-Aldrich), 1 µM dexamethasone (Sigma-Aldrich), 100 ng/ml cholera toxin (Sigma-Aldrich), 5 ml/l insulin-transferrin-selenium (Corning), and 100 U/ml penicillin/streptomycin (Gibco/Thermo Scientific). Collagenase I was from MilliporeSigma (St. Louis, MO). Rat tail collagen I was from BD Biosciences (San Diego, CA). All antibodies used for western blotting, immunohistochemistry and immunofluorescence were from the following sources: Peprotech (Rocky Hill, NJ), Abcam (Cambridge, MA), Cell Signaling Technologies (Danvers, MA), Dako (Santa Clara, CA), STEMCELL Technologies (Vancouver, Canada), Biorad (Hercules, CA), and LifeSpan BioSciences (Seattle, WA), BD Biosciences (Franklin Lakes, NJ), BioLegend (San Diego, CA), GeneTex (Irvine, CA), Miltenyi Biotec (Auburn, CA) and are described in detail in **Key Resources Table**. All neutralizing antibodies that have been used in animal studies are described in detail in Materials and methods section for these experiments as well as in **Key Resources Table**. Secondary HRP-linked anti-mouse or anti-rabbit antibodies were from Jackson ImmunoResearch (West Grove, PA). DAPI and LPS were from Sigma-Aldrich. IL4, IFNγ, and CXCL10 were from PeproTech (Rocky Hill, NJ).

## Isolation of primary macrophages (peritoneal or bone marrow derived) and polarization

Peritoneal macrophages: Non-transgenic mice were injected with 2–3 ml of 5% aged thioglycollate. Five days later, mice were euthanized and peritoneal macrophages were collected immediately. A small incision was made in the peritoneum and RPMI-1640 media with 10% FBS was flushed through the peritoneal cavity using a sterile transfer pipette. The peritoneal wash was centrifuged, washed with fresh RPMI1640 + 10% FBS media, and cells were plated onto 10 cm dishes. After 2–3 hr, once macrophages completely adhered to the plates, cells were washed two to three times to remove non-adherent cells and fresh media was added. Cells were allowed to acclimatize overnight before harvesting and experimental manipulation. Bone marrow-derived macrophages (BMDM): Femurs of non-transgenic mice were cut proximal to the joints on both ends and washed twice in cold, sterile PBS (without calcium and magnesium) to remove excess blood and muscle. Using forceps, the bones were held over a sterile 50 ml tube, and the bone cavity was flushed twice with 5 ml cold PBS using a 25G needle and syringe. The bones were discarded, and the bone marrow was pelleted (10 min, 500 × g, RT), then gently resuspended to single cells in 10 ml warm macrophage complete medium (DMEM-F12 with L-glutamine + 10% FBS + 100 U/ml penicillin + 100 µg/ml streptomycin + 100 U/ml of M-CSF) (Zhang et al., 2008). After counting, $4 \times 10^5$ cells were added to 10 cm dishes in 10 ml macrophage complete medium. On days 3 and 5, 5 ml of medium was replaced with fresh complete medium. On day 7, the cells were polarized. Polarization of peritoneal or bone marrow-derived macrophages to M1 or M2 phenotypes was obtained by stimulation with LPS + IFNγ (M1 polarization) and/or with IL-4 (M2 polarization) for 24 hr as previously described (Liou et al., 2017).

## Isolation of pancreas tissue resident macrophages

Pancreata from 8 week old, non-transgenic mice were harvested by washing the pancreas in HBSS, mincing the tissue, centrifuging (931 × g, 4°C, 2 min), and then dissociating the tissue in collagenase (2 mg/ml, 5 ml, 37°C, 20 min, 220 rpm). To stop dissociation, HBSS + 5% FBS was added to the pancreas–collagenase mixture and washed two additional times with HBSS + 5% FBS (931 × g, 4°C, 2 min). Pancreas cells were then filtered through 500 µm and 105 µm meshes prior to adding the cell suspension to HBSS + 30% FBS and centrifuging (233 × g, 4°C, 2 min) to obtain a cell pellet. Cells were then labeled with F4/80 magnetic beads (Miltenyi Biotec, Auburn, CA) to isolate pancreas-resident macrophages. Cells were filtered through a 40 µm mesh to obtain a single-cell suspension, and then cells were incubated with F4/80 magnetic beads in MACS buffer (15 min, 4°C, as per the manufacturer's instructions) (MACS buffer: PBS, 0.5% BSA, 2 mM EDTA). Cells were washed with MACS buffer (300 × g, 4°C, 10 min), filtered through 40 µm mesh, and then applied to pre-washed LS Columns (Miltenyi Biotec, Auburn, CA). Per column instructions, magnetically labeled cells were acquired and plated in DMEM-F12 + 10% FBS + 1% L-glutamine + penicillin/streptomycin + 0.1 µg/ml macrophage colony stimulating factor (Peprotech, Rocky Hill, NJ).

## Genetic animal model and treatments

*Ptf1a/p48cre/+* and *LSL-KrasG12D/+* mouse strains and genotyping of mice have been described previously (Liou et al., 2015). Seven to 8 week old *Ptf1a/p48cre;LSL-KrasG12D* (KC) or non-transgenic (ntg) mice with the same background were injected intraperitoneally (IP) with a CXCR3 neutralizing antibody (CXCR3 NAB; BE0249) (Bio X Cell, West Lebanon, NH) or an isotype control IgG; BE0091 (Bio X Cell) at 200 µg/mouse for 9 weeks. Males and females were randomly allocated to different groups since there are no sex-based differences observed in this model. All animal experiments were conducted under IACUC-approved protocols (A50214-14-R17, A30615-15-R18) and were run in accordance with institutional guidance and regulation.

For T-cell depletion, 6 week old *Ptf1a/p48cre;LSL-KrasG12D* mice were intraperitoneally injected with both anti-mouse CD4 (BP0003, Bio X Cell) and anti-mouse CD8α (BP0061, Bio X Cell) antibodies, or their IgG2b isotype control (BP0090, Bio X Cell). Two hundred micrograms of each antibody was injected per mouse for five consecutive days. After T-cell depletion, mice were segregated into different groups and injected with CXCR3 NAB (BE0249, Bio X Cell) or IgG isotype control antibodies (BE0091, Bio X Cell).

## Human pancreatic tissue samples

Patient tissues were obtained from archival materials in accordance with institutional guidelines and prior institutional review board (IRB) approval.

## Immunofluorescence on cells and tissue

For paraffin-embedded tissue, slide sections were deparaffinized, and antigen retrieval was performed with 10 mM sodium citrate buffer (pH 6.0) for 25 min at 100°C. Then, they were treated with 3% $H_2O_2$ for 15 min at room temperature, rinsed with PBS, and blocked with serum-free Protein Block (DAKO, Santa Clara, CA) for 1 hr at room temperature. Slides were incubated with primary antibodies diluted in 150 µl per slide of Antibody Diluent (Dako) overnight. Slides were washed three times with 0.5% Tween20 in PBS and incubated with appropriate Alexa-Fluor labeled secondary antibodies (Invitrogen, Carlsbad, CA) at 1:500 and DAPI at 125 µg/ml for 1 hr. Slides were washed three times with 0.5% Tween20 in PBS followed by cover-slipping with PermaFluor mounting media (Thermo Fisher Scientific, Waltham, MA). For imaging, whole slides were scanned using the Aperio fluorescence scanner (Leica, Buffalo Grove, IL) or the Pannoramic 250 Flash III (3DHISTECH).

For cellular staining, cells grown in ibidi chamber slides were washed with cold PBS and fixed with 4% paraformaldehyde (PFA) for 15 min at 37°C. Fixed cells were washed with PBS and permeabilized with 0.1% Triton X-100 for 10 min at room temperature. Then, cells were blocked with 3% BSA + 0.05% Tween in PBS for 30 min at room temperature. Incubation with primary antibodies was done overnight at 4°C. Cells were washed with PBS and incubated with secondary antibodies at 1:500 and DAPI at 125 µg/ml for 2 hr. Cells were washed, and mounting media was added to the wells, followed by imaging on a fluorescence microscope (Carl Zeiss, Thornwood, NY).

## Flow cytometry

Murine pancreata were resected, and tissue was dissociated using the mouse tumor dissociation kit (Miltenyi, Bergisch Gladbach, Germany) per the manufacturer's protocol. Dissociated tissue was sequentially passed through 500 µm (Repligen, Boston, MA), 105 µm (Repligen), and 40 µm (Thermo Fisher Scientific) filters to acquire a single-cell suspension before magnetic isolation of $CD45^+$ cells. Cells were incubated with $CD45^+$ MicroBeads (Miltenyi) followed by magnetic separation using LS Columns (Miltenyi) on a QuadroMACS magnet (Miltenyi) per the manufacturer's protocol. $CD45^+$ cells were then labeled with LIVE/DEAD Fixable Violet Dead Cell Stain Kit (Thermo Fisher Scientific) before subsequent fixation and staining using the PerFix-nc kit (Beckman Coulter, Brea, CA) as per the manufacturer's protocol. Antibodies are described in detail in **Key Resources Table**. The Attune NxT (Thermo Fisher Scientific) was used for multicolor flow cytometric detection, and analysis was done using FlowJo (BD Biosciences).

## In situ hybridization

The procedure for ISH has been described in detail previously (*Bastea et al., 2019*). Briefly, ISH was performed using RNAscope Assay 2.5 HD Reagent Kit–Brown or RNAscope Multiplex Fluorescent Reagent Kit v2 (Advanced Cell Diagnostics, Hayward, CA). Formalin-fixed, paraffin-embedded (FFPE) sections (5 µm) were baked at 60°C for 1 hr, deparaffinized in xylene for 15 min, dehydrated in 100% ethanol, and dried at room temperature overnight in a desiccator. Next day, slides were treated with hydrogen peroxide for 10 min, followed by target retrieval for 8 min. Protease treatment was done for 15 min at 40°C in a hybridization oven. Next, slides were incubated with the appropriate probe for 2 hr. RNAscope target probes used were *Cxcl10* (Mm-Cxcl10 408921), *Cxcr3* (Mm-Cxcr3 4025110), *CXCR3* (Hs-CXCR3 539251), and *Ifng* (Mm-Ifng 311391). After probe hybridization, amplification steps were followed according to the manufacturer's protocol, except Amp 5 step, which was modified to 1 hr incubation in the DAB procedure (RNAscope Assay 2.5 HD Reagent Kit–Brown). Slides were counterstained with hematoxylin, dehydrated, and mounted.

When followed by IF, slides were treated with boiling 10 mM citrate buffer (pH 6.0) for 10 min to repeat antigen retrieval and the regular IF protocol was followed. Primary antibodies were used at concentrations listed in **Key Resources Table**.

For the fluorescent detection (RNAscope Multiplex Fluorescent Reagent Kit v2), following AMP1, 2 and 3 steps according to the manufacturer's protocol, probes were labeled using the Opal 690 fluorophore reagent pack (Perkin Elmer, Waltham, MA). Next, regular IF procedure was performed for

co-staining. Samples were counterstained with DAPI and mounted. For imaging, whole slides were scanned at 40× magnification using the Aperio fluorescence scanner (Leica).

## Isolation of primary acinar cells

Pancreas was removed quickly following euthanasia and placed in cold HBSS media. After washing in cold HBSS media twice, pancreas was minced into small 1–5 mm pieces and digested with collagenase I in a 37°C shaker for 20 min. After 20 min, the digestion was stopped by adding an equal volume of cold HBSS media with 5% FBS. The digested pieces were passed through a 500 µm mesh. More 5% FBS media was passed through the mesh with digested pieces to further strain them. The filtrate was then passed through a 105 µm mesh. The filtrate from this step was added dropwise to a tube containing HBSS media with 30% FBS. The cell suspension was centrifuged at 205 × g for 2 min at 4°C. Acinar cells were then resuspended in complete Waymouth's media (1% FBS, 0.1 mg/ml soybean trypsin inhibitor, 1 µg/ml dexamethasone).

## 3D collagen explant culture of pancreatic acinar cells

Cell culture plates were coated with collagen I in Waymouth media without supplements. Freshly isolated primary pancreatic acinar cells were added as a mixture with collagen I/Waymouth media on the top of this layer (3D on-top method). Furthermore, Waymouth complete media was added on top of the cell/gel mixture and replaced the following day and then every other day. When inhibitors, peptides, or proteins were added, the compound of interest was added to both, the cell/gel mixture and the media on top. To express proteins using adenovirus, acinar cells were infected with adenovirus of interest and incubated for 3–5 hr before embedding in the collagen I/Waymouth media mixture. At day 6 or 7 (dependent on time course of duct formation), numbers of ducts were counted under a microscope, and photos were taken to document structures.

## Adenoviral infection of acinar cells

For adenoviral infection, Adeno-cre-GFP or Adeno-null-GFP viruses (Vector Biolabs, Malvern, PA) were added to the resuspended acinar cells in Waymouth's complete media in a non-adherent dish. Infection was carried out for 3 hr with gentle swirling every 15 min during the first hour. After the 3 hr infection, cell suspension was mixed with Rat tail collagen-Type I (BD Biosciences, San Jose, CA) and plated.

## Lentiviral infection of PanIN lesion cells, organoid formation, and orthotopic implantation in athymic nude mice

SM3 PanIN cells were infected using control lenti-eGFP (LPP-EGFP-Lv105-025-C) or lenti-cxcl10 (LPP-Mm03214-Lv105-100) lentiviral particles (Genecopoeia, Rockville, MD) with 5 µg/ml of polybrene (Santa Cruz Biotechnology, Dallas, TX) at 70–80% confluency. Infected cells were incubated at 4°C for 2 hr and then at 37°C overnight. On the following day, media containing lentiviral particles and polybrene was removed and replaced with fresh media to allow the cells to recover. Next day, selection media containing 2.5 µg/ml puromycin was added to the cells and was replaced every 2–3 days. After 11 days of selection, cells were harvested and embedded in Matrigel (Corning, Corning, NY) with regular media without puromycin. After 2 days, cells formed duct-like PanIN organoid structures. Using a non-enzymatic cell dissociation reagent (Corning), PanIN organoids were harvested and mixed with activated primary pancreatic stellate cells at a ratio of 1:4. Cell mixture was resuspended in phenol-red free Matrigel, and 50 µl (25,000 organoid cells + 100,000 activated primary stellate cells) was injected directly into the pancreas of athymic nude ($Foxn1^{nu}$) mice (Jackson laboratory, Bar Harbor, ME). Wound was closed using 4–0 Vicryl sutures and 7 mm wound clips. After 1 week of recovery, wound clips were removed, and ultrasound imaging was performed to ensure implantation of cells.

## RNA extraction and quantitative PCR

Cells were washed with cold PBS, and RNA extraction was done using the RNeasy PLUS Mini Kit (Qiagen, Germantown, MD). cDNA was prepared using the High Capacity cDNA RT Kit (Applied Biosystems, Foster City, CA). TaqMan Fast Mix 2x (Applied Biosystems) was used to prepare qPCR along with the primer/probe sets described in **Key Resources Table.** Reactions were run on the

QuantStudio 7 Flex Real-Time PCR System (Applied Biosystems). All $C_T$ values were normalized to Gapdh, and the $\Delta\Delta C_T$ method was used to calculate fold changes.

## Protein isolation and western blot

Cells were washed with cold PBS (140 mM NaCl, 2.7 mM KCl, 8 mM $Na_2HPO_4$, and 1.5 mM $KH_2PO_4$ [pH 7.2]) twice and lysed using RIPA buffer with a protease inhibitor cocktail (Sigma-Aldrich). Lysates were incubated on ice for 30 min followed by centrifugation at 13,000 rpm for 15 min at 4°C on a table top centrifuge. Protein supernatant was separated from cell debris, and concentration was measured using the BioRad Protein Assay (BioRad, Hercules, CA). Samples were run on SDS-PAGE gels and transferred to nitrocellulose membranes. Proteins of interest were detected using the appropriate primary antibodies at indicated concentrations (**Key Resources Table**) and horseradish peroxidase (HRP)-conjugated secondary antibodies.

## Transwell chemoattraction assays

Peritoneal or bone marrow-derived macrophages were seeded in serum-free RPMI media on 5 μm transwell permeable inserts (Corning, Corning, NY). Six hundred microliters of control media or media containing 500 ng/ml of CXCL10 was placed in the bottom wells. Experiments were conducted in triplicates for each condition. Cells were incubated for the indicated time. Inserts were then carefully removed, washed with PBS, and fixed in 4% PFA for 15 min at 37°C. After gently washing the inserts twice, cells were permeabilized with 0.1% Triton X-100 for 10 min at room temperature and stained with DAPI for counting.

## MTT proliferation assay

Proliferation of pancreas-resident macrophages was analyzed via MTT assay (Sigma, St. Louis, MO), where wells were incubated with MTT labeling reagent (4 hr, 37°C, 5% $CO_2$). Following MTT labeling, cells were incubated with solubilization solution (10% SDS in 0.01 M HCl) overnight (37°C, 5% $CO_2$), and absorbance was read the following morning at 500 nm (Synergy HT plate reader, BioTek, Winooski, VT).

## Cytokine array

The cytokines secreted by SM3 cells were determined using the Proteome Profiler Mouse Cytokine Array Kit (R and D Systems) according to the manufacturer's instructions.

## Quantification and statistical analysis

All cell biological and biochemical experiments have been performed at least three times. For animal experiments, if not stated otherwise in the figure legends, pancreatic samples from n = 3 mice have been used for quantification analyses. IHC data was quantified by manual counting of positive cells or by using the Aperio Positive Pixel Count Algorithm. Data are presented as mean ± standard deviation (SD). If not stated otherwise in the figure legends, p-values were acquired with the unpaired student's t-test with Welch's correction using Graph Pad software (GraphPad Inc, La Jolla, CA). For all experiments in which pancreatic areas were compared, we transformed the data via *arcsin* transformation (ASIN(SQRT(proportion of abnormal tissue area to total tissue area))*180/PI) before a t-test was performed. $p < 0.05$ was considered statistically significant.

## Acknowledgements

We thank Dr. Yan Asmann (Associate Professor of BioMedical Informatics, Mayo Clinic) for advice on the statistical analyses. We also thank our colleagues in the Storz laboratory and Department for Cancer Biology for helpful discussions. We also thank the Champions For Hope (Funk-Zitiello Foundation) and Gary Chartrand Foundation for their continuous support.

# Additional information

## Funding

| Funder | Grant reference number | Author |
|---|---|---|
| National Cancer Institute | CA229560 | Peter Storz |
| National Cancer Institute | CA200572 | Peter Storz |
| National Cancer Institute | P50CA102701 | Peter Storz |

The funders had no role in study design, data collection and interpretation, or the decision to submit the work for publication.

## Author contributions

Veethika Pandey, Conceptualization, Data curation, Formal analysis, Validation, Visualization, Writing - original draft, Writing - review and editing; Alicia Fleming-Martinez, Data curation, Formal analysis, Validation, Investigation, Methodology, Writing - review and editing; Ligia Bastea, Data curation, Formal analysis, Validation, Investigation, Writing - review and editing; Heike R Doeppler, Data curation, Formal analysis, Validation, Investigation; Jillian Eisenhauer, Data curation, Formal analysis; Tam Le, Data curation; Brandy Edenfield, Data curation, Visualization; Peter Storz, Conceptualization, Resources, Funding acquisition, Validation, Investigation, Writing - original draft, Project administration, Writing - review and editing

## Author ORCIDs

Peter Storz (iD) https://orcid.org/0000-0002-0132-5128

## Ethics

Animal experimentation: This study was performed in strict accordance with the recommendations in the Guide for the Care and Use of Laboratory Animals of the National Institutes of Health. All animal experiments were conducted under IACUC approved protocols (A50214-14-R17, A30615-15-R18) and were run in accordance with institutional guidance and regulation.

## Decision letter and Author response

Decision letter https://doi.org/10.7554/eLife.60646.sa1
Author response https://doi.org/10.7554/eLife.60646.sa2

# Additional files

## Supplementary files

• Transparent reporting form

## Data availability

All data generated or analysed during this study are included in the manuscript and supporting files. Source data have been included for all Figures and Figure supplements.

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

# Appendix 1

**Appendix 1—key resources table**

| Reagent type (species) or resource | Designation | Source or reference | Identifiers | Additional information |
|---|---|---|---|---|
| Antibody | anti-CD3 (Rabbit polyclonal) | Abcam | Cat# ab5690, RRID: AB_305055 | IHC (1:400), IF (1:400) |
| Antibody | Anti-CD3 (Rat monoclonal) | Abcam | Cat# ab11089, RRID:AB_2889189 | IF (1:200) |
| Antibody | anti-CD3 (Rat monoclonal) | BioLegend | Cat# 100222, RRID: AB_2242784 | FC (0.2 µg/1 × $10^6$ cells) |
| Antibody | anti-CD4 (Rabbit monoclonal) | Abcam | Cat# ab183685, RRID:AB_2686917 | IF (1:1000) |
| Antibody | anti-CD4 (Rat monoclonal) | BioLegend | Cat# 100456, RRID: AB_2565845 | FC (0.2 µg/1 × $10^6$ cells) |
| Antibody | anti-CD45 (Rat monoclonal) | BioLegend | Cat# 103137, RRID: AB_2561392 | FC (0.3 µg/1 × $10^6$ cells) |
| Antibody | anti-CD45 Microbeads (Rat monoclonal) | Miltenyi Biotec | Cat# 130-052-301, RRID:AB_2877061 | MACS (10 µL/1 × $10^7$ cells) |
| Antibody | anti-CD68 (Mouse monoclonal) | DAKO/Agilent | Cat# M0876, RRID: AB_2074844 | IF (1:100) |
| Antibody | anti-CD8 alpha (Rabbit monoclonal) | Abcam | Cat# ab209775, RRID:AB_2860566 | IF (1:500) |
| Antibody | anti-CD8a (Rat monoclonal) | BioLegend | Cat# 100747, RRID: AB_11219594 | FC (0.2 µg/1 × $10^6$ cells) |
| Antibody | anti-CD80 (Armenian hamster monoclonal) | BioLegend | Cat# 104705, RRID: AB_313126 | FC (1 µg/1 × $10^6$ cells) |
| Antibody | anti-CXCL10 (Rabbit polyclonal) | PeproTech | Cat# 500-P129bt-50ug, RRID:AB_148105 | WB (1:2000) |
| Antibody | anti-CXCR3 (Rabbit polyclonal) | LifeSpan Biosciences | Cat# LS-C332293, RRID:AB_2891301 | IF (1:50) |
| Antibody | anti-CXCR3 (Armenian hamster monoclonal) | BD Biosciences | Cat# 742274, RRID: AB_2871450 | FC (0.2 µg/1 × $10^6$ cells) |
| Antibody | anti-F4/80 (Rat monoclonal) | Bio-Rad | Cat# MCA497R, RRID:AB_323279 | IHC (1:250), IF (1:250) |
| Antibody | anti-F4/80 (Rat monoclonal) | BioLegend | Cat# 123133, RRID: AB_2562305 | FC (0.3 µg/1 × $10^6$ cells) |
| Antibody | anti-F4/80 MicroBeads (Mouse monclonal) | Miltenyi Biotec | Cat# 130-110-443, RRID:AB_2858241 | MACS (10 µl/1 × $10^7$ cells) |
| Antibody | anti-GAPDH (Rabbit monoclonal) | Cell Signaling Technology | Cat# 5174, RRID: AB_10622025 | WB (1:1000) |
| Antibody | anti-iNOS (Rabbit polyclonal) | Abcam | Cat# ab3523, RRID: AB_303872 | IF (1:200) |
| Antibody | anti-iNOS (Mouse monoclonal) | Abcam | Cat# ab49999, RRID:AB_881438 | IF (1:100) |
| Antibody | anti-NKG2D (Rabbit polyclonal) | GeneTex | Cat# GTX50988, RRID:AB_2891302 | IF (1:200) |
| Antibody | anti-SMA (Rabbit polyclonal) | Abcam | Cat# ab5694, RRID: AB_2223021 | IHC (1:200) |
| Antibody | anti-pY641-STAT6 (Rabbit monoclonal) | Cell Signaling Technology | Cat# 56554, RRID: AB_2799514 | IF (1:400) |

*Continued on next page*

*Appendix 1—key resources table continued*

| Reagent type (species) or resource | Designation | Source or reference | Identifiers | Additional information |
|---|---|---|---|---|
| Antibody | anti-pY701-STAT1 (Rabbit monoclonal) | Cell Signaling Technology | Cat# 9167, RRID: AB_561284 | IF (1:400) |
| Antibody | anti-pY701-STAT1 (Mouse monoclonal) | Abcam | Cat# ab29045, RRID:AB_778096 | IF (1:200), WB (1:1000) |
| Antibody | anti-STAT1 (Rabbit polyclonal) | Cell Signaling Technology | Cat# 9172, RRID: AB_2198300 | WB (1:1000) |
| Antibody | anti-YM1 (Rabbit polyclonal) | STEMCELL Technologies | Cat# 60130, RRID: AB_2868482 | IF (1:200) |
| Antibody | anti-YM1/2 (Rabbit monoclonal) | Abcam | Cat# ab205491, RRID:AB_2891303 | FC (1 µg/1 × $10^6$ cells), IF (1:100) |
| Antibody | CXCL10 neutralizing antibody (NAB; Rat monoclonal) | R and D Systems | Cat# MAB466, RRID:AB_2292486 | CXCL10 NAB |
| Antibody | Isotype control IgG2A antibody (Rat monoclonal) | R and D Systems | Cat# MAB006, RRID:AB_357349 | CXCL10 NAB control |
| Antibody | CXCR3 neutralizing antibody (NAB; Armenian hamster monoclonal) | Bio X Cell | Cat# BE0249, RRID: AB_2687730 | CXCR3 NAB |
| Antibody | Isotype control IgG antibody (Armenian hamster polyclonal) | Bio X Cell | Cat# BE0091, RRID: AB_1107773 | CXCR3 NAB control |
| Antibody | anti-CD4 (Rat monoclonal) | Bio X Cell | Cat# BP0003-1, RRID:AB_2891358 | T-cell depletion |
| Antibody | anti-CD8α (Rat monoclonal) | Bio X Cell | Cat# BP0061, RRID: AB_2891359 | T-cell depletion |
| Antibody | Isotype control IgG2b antibody (Rat monoclonal) | Bio X Cell | Cat# BP0090, RRID: AB_2891360 | T-cell depletion control |
| Cell line (*Mus musculus*) | SM3 | *Agbunag et al., 2006*; *Liou et al., 2017* | | Primary duct-like cells from KC mouse |
| Chemical compound, drug | NVP-BSK805 | Selleckchem | Cat# S2686 | pan-JAK inhibitor |
| Commercial assay or kit | Mouse tumor dissociation kit | Miltenyi Biotec | Cat# 130-096-730 | |
| Commercial assay or kit | RNAscope Assay 2.5 HD Reagent Kit- Brown | Advanced Cell Diagnostics | | In situ hybridization (brown) |
| Commercial assay or kit | RNAscope Multiplex Fluorescent Reagent Kit v2 | Advanced Cell Diagnostics | | In situ hybridization (fluorescent) |
| Peptide, recombinant protein | Recombinant murine CXCL10 | Peprotech | Cat# 250–16 | |
| Peptide, recombinant protein | Recombinant murine IL-4 | Peprotech | Cat# 214–14 | |
| Peptide, recombinant protein | Recombinant murine IFNɣ | Peprotech | Cat# 315–05 | |
| Sequence-based reagent | *Arg1* | TaqMan (Thermo Fisher Scientific) | Mm00475988_m1 | qPCR probe |
| Sequence-based reagent | *Chil3* | TaqMan (Thermo Fisher Scientific) | Mm00657889_mH | qPCR probe |
| Sequence-based reagent | *Cxcl10* | TaqMan (Thermo Fisher Scientific) | Mm00445235_m1 | qPCR probe |

*Appendix 1—key resources table continued*

| Reagent type (species) or resource | Designation | Source or reference | Identifiers | Additional information |
|---|---|---|---|---|
| Sequence-based reagent | *Cxcr3* | TaqMan (Thermo Fisher Scientific) | Mm99999054_s1 | qPCR probe |
| Sequence-based reagent | *Irf4* | TaqMan (Thermo Fisher Scientific) | Mm00516431_m1 | qPCR probe |
| Sequence-based reagent | *Irf5* | TaqMan (Thermo Fisher Scientific) | Mm00496477_m1 | qPCR probe |
| Sequence-based reagent | *Gapdh* | TaqMan (Thermo Fisher Scientific) | Mm99999915_g1 | qPCR probe |
| Sequence-based reagent | *Retnla* | TaqMan (Thermo Fisher Scientific) | Mm0045109_m1 | qPCR probe |
| Software, algorithm | Aperio ImageScope | Leica Biosystems | | Tissue analysis |
| Software, algorithm | Aperio ImageScope Positive Pixel Algorithm | Leica Biosystems | | Tissue analysis |
| Software, algorithm | FlowJo | BD Biosciences | | Flow cytometry analysis |
| Software, algorithm | GraphPad | GraphPad, Inc | | Statistical analysis |
| Other | Aperio AT2 Digital Scanner | Leica Biosystems | | Brightfield tissue scans |
| Other | Aperio FL Slide Scanner | Leica Biosystems | | Fluorescent tissue scans |
| Other | Pannoramic 250 Flash III | 3DHISTECH | | Fluorescent tissue scans |
| Other | Attune NxT Flow Cytometer | Thermo Fisher Scientific | | |
| Other | *Cxcl10* ISH probe (mouse) | Advanced Cell Diagnostics | Cat# 408921 | In situ hybridization probe |
| Other | *Cxcr3* ISH probe (mouse) | Advanced Cell Diagnostics | Cat# 402511 | In situ hybridization probe |
| Other | *Cxcr3* ISH probe (human) | Advanced Cell Diagnostics | Cat#539251 | In situ hybridization probe |
| Other | *Ifng* ISH probe (mouse) | Advanced Cell Diagnostics | Cat# 311391 | In situ hybridization probe |
| Other | Lipopolysaccharides (LPS) | Sigma-Aldrich | Cat# L4391 | Lipopolysaccharides from *Escherichia coli* O111:B4; γ-irradiated; suitable for cell culture |
| Other | Adeno-Cre-GFP | Vector Biolabs | Cat# 1700 | Adenovirus |
| Other | Adeno-Null-GFP | Vector Biolabs | Cat# 1300 | Adenovirus |
| Other | CXCL10 | GeneCopoeia | LPP-EGFP-Lv105-025-C | Lentiviral particles |
| Other | eGFP (control) | GeneCopoeia | LPP-Mm03214-Lv105-100 | Lentiviral particles |
| Other | Proteome Profiler Mouse Cytokine Array Kit | R and D Systems | Cat# ARY006 | Cytokine array |

