## [Decision Letter]

**Acceptance summary:**

This manuscript reports a role for the CXCL10/CXCR3 signaling axis in pancreatic preneoplastic lesions in promoting inflammatory macrophages phenotypes. The authors provide data to support a causal role of CXCL10 and CXCR3 in the recruitment and polarization of macrophages in the pancreas, as well as the degree of fibrosis in PanIN lesions. The work provides insight into mechanisms that affect tumor formation.

**Decision letter after peer review:**

Thank you for submitting your article "CXCL10/CXCR3 contributes to an inflammatory microenvironment and its blockade enhances pancreatic cancer development" for consideration by *eLife*. Your article has been reviewed by 3 peer reviewers, and the evaluation has been overseen by a Reviewing Editor and Tadatsugu Taniguchi as the Senior Editor. The following individual involved in review of your submission has agreed to reveal their identity: Mara Sherman (Reviewer #3).

The reviewers have discussed the reviews with one another and the Reviewing Editor has drafted this decision to help you prepare a revised submission.

This manuscript reports a role for the CXCL10/CXCR3 signaling axis in pancreatic preneoplastic lesions in promoting inflammatory macrophages phenotypes. The authors provide data to support a causal role of CXCL10 and CXCR3 in the recruitment and polarization of macrophages in the pancreas, as well as the degree of fibrosis in PanIN lesions. The work provides potential insight into mechanisms that affect tumor formation.

All reviewers agreed that the data were interesting and potentially appropriate for publication in *eLife*. With that said, the reviewers were also in agreement that some conclusions were overstated, and the data presentation needs to be improved included more attention to controls and statistical analysis. I have included all comments from the reviewers, however when submitted a revised manuscript it is essential to address the following points:

1. Please clarify the statistical methods used and provide information on replicates for all data.

2. The reviewers all agreed that the CXCL10 data comprises the core of the findings, but though those findings could be strengthened with further experiments. The current *eLife* policy also allows limiting the claims if new experiments are not possible. Specifically, to support the claim that CXCL10/CXCR3 blockade enhances cancer development, one suggested additional experiment is to evaluate tumor formation in KC mice upon longer treatment with CXCR3 Nab. However, this experiment would take a long time and thus may not be feasible. In the absence of further support that CXCL10/CXCR3 blockade enhances cancer development, and alternative would be to focus your conclusions on the role of CXCL10 in promoting an inflammatory macrophage identity. This latter option likely would include adjusting the title of the manuscript which refers to a role in tumor development.

3. Please address the questions raised by reviewers around the source of the macrophages as this speaks to the central claims of the study.

4. Address overstatements suggested by the reviewers.

5. Some reviewers questioned how much the IRF4/5 data added to the paper? One possibility is to focus entirely on the role of CXCL10 in promoting an inflammatory macrophage identity, others thought this was interesting but perhaps is better included as supplementary data.*Reviewer #1 (Recommendations for the authors):*

General assessment of the work:

Overall, the manuscript addresses one piece of a major question in the pancreatic cancer field – how the inflammatory microenvironment of these tumors is sculpted through paracrine signaling pathways. In general, while the data reported are supportive of their conclusions, in most cases alternative explanations were not ruled out. This could be addressed with additional experiments. In particular, the question of the initial trigger for the CXCL10/CXCR3 pathway did not exclude other explanations and this was a major component of the story. In addition, there were a number of technical concerns and questions that limit our ability to interpret the data. These could be remedied in a resubmission but require careful attention to detail.

Substantive concerns:

1. Utilized ISH data as proof for protein expression. While necessary in order to identify which cell synthesize secreted proteins like CXCL10, it would be informative to see paired protein staining or Western blots for those experiments that don't already include them, as mRNA expression isn't always indicative of protein expression/levels.

2. In figure 1, the authors conclude that IFNγ is responsible for activation fo CXCL10 expression. Treatment with IFNγ did induce CXCL10, but this only proves that the pathway is sufficient and does not address whether it is necessary. To learn this, one could use an anti-IFNγ antibody and see if this blocks CXCL10 activation. Otherwise, the conclusions drawn from this experiment should be stated in a more constrained manner.

3. Similarly, the conclusions drawn regarding T cell and NK cell involvement in the CXCL10/CXCR3 pathway are not fully supported.

– The combined ISH/IF experiment in Figure 1G is an elegant technical approach to studying the source-cell of a secreted protein. However, not a single control was presented for this experiment, making it impossible to interpret these results.

– Presence of T cells and NK cells in the area around precursor cells, and the knowledge that T cells and NK cells express IFNγ is not sufficient to conclude that T cells and NK cells are responsible for initiation of CXCL10 expression. It is a potential explanation, but no work was done to rule out other unknown factors.

– Additionally, when done in a T cell free environment, the pathway still functioned and the authors themselves claim that there is "no significant role of T cells in CXCL10/CXCR3 driven development of pancreatic cancer." IFNγ from T cells cannot be both the initial driver and also not significant.

4. In Figure 2B, was the migration assay performed on M2 macrophages? One would expect them not to be responsive based on 2A, but this should be examined.

5. The conclusion that macrophage polarization occurred through the IRF4/5 transition is not sufficiently supported. It is not clear that the reported observation should even be included is it is not integral to the overall message of the paper.

6. Please provide a statistical analysis of the data in 4G, particularly with regard to the incidence of higher grade lesions. Given the variance in the extent of PanIN lesions that present in the KC model, it is generally necessary to analyze far more than 4 animals per group to assess impact on this type of phenotype.

7. In general, for many experiments it was not clear how many replicates were performed and whether these were technical or biological replicates. It is important that this be stated the legend for each panel in each figure, otherwise it is impossible to provide an informed analysis of the data.

8. Related to the prior point, the data representation in most figures utilized outdated approaches (eg. bar plots) that communicate a minimum amount of information about distribution, variance, and replicates. We strongly recommend the adoption of more modern, informative data display conventions, such as those described in PMID: 32346721.

9. Additionally, it is important to indicate which statistical test was utilized in each figure. The blanket statement in the Methods that Student's T was used is not sufficient as this is not always the appropriate test. For example, in Figure 4B, the authors present normalized ratiometric data (% positive area). A Student's T is not an appropriate test for this comparison because ratiometric data violate the assumption of a normal distribution required by this test. We strongly suggest the authors review every experiment with a biostatistician.

*Reviewer #2 (Recommendations for the authors):*

CXCL10/CXCR3 contributes to an inflammatory microenvironment and its blockade enhances pancreatic cancer development

In this manuscript, Pandey et al., reports a novel CXCL10/CXCR3 signaling axis between pancreatic preneoplastic lesions and inflammatory macrophages to shape the microenvironment of PanIN. The authors provide strong in vivo experimental data to support a causal role of CXCL10 and CXCR3 in the recruitment and polarization of macrophages in the pancreas, as well as the degree of fibrosis in PanIN lesions. Addressing the following concerns will further strengthen the manuscript:

1. CXCL10 has previously been reported to be the major chemokine produced by pancreatic stellate cells (PSCs) in the PDA tumor microenvironment. CAFs expressing IP10 are thought to recruit Tregs to promote immune evasion and tumorigenesis. CXCL10 levels have also been shown to positively correlate with stromal content in human PDAC (PMID25415223). In the current manuscript, the authors report a very robust increase in fibrosis in PanIN (using SMA as a marker) when the CXCL10/CXCR3 axis is supressed using neutralizing antibodies. This raises the interesting question of whether CXCL10 exhibit opposite functions during premalignant vs malignant stages of tumorigenesis. This can be addressed using a syngeneic orthotopic model of PDA to evaluate the effect of CXCL10/CXCR3 inhibition in established tumors. Alternatively, the authors may consider revising the title of their manuscript to focus on the effect of CXCL10/CXCR3 in preneoplastic lesions as opposed to pancreatic cancer. CXCR3 Nab did not lead to a significant change in % of PanIN or ADM in mice (Figure 4G) so it is hard to extrapolate from the current data what the role of this signaling axis is in the context of cancer unless additional data is provided beyond the current timepoints to evaluate tumorigenesis.

2. What is the spatial relationship between interferon-γ-producing T cells and PanIN lesions? A multiplex FISH of CXCL10 and IFNγ will strengthen the model that short-distance-acting IFNγ from T cells within the pancreatic parenchyma induces CXCL10 production in neighbouring acinar/ADM cells. Along the same lines, what is the percentage of T cells in PanIN and ADM-bearing pancreata?

3. Tumor associated macrophages are generally thought to originate from circulating monocytes derived from bone marrow HSCs. While thioglycollate activated peritoneal macrophages is a tractable experimental model, they are phenotypically very different from non-elicited, resident peritoneal macrophages, or naïve bone marrow derived macrophages, and as such do not serve as a proper control for "non-polarized" macrophages. Experiments presented in Figure 2A and B should also be done using naïve (M0) and M1 , M2 macrophages derived from the bone marrow. Experimental evidence showing that CXCR3-expressing macrophages in the pancreas originated from the peritoneum is currently absent, the authors should therefore refrain from drawing conclusions on the origin of these macrophages.

4. The authors should extend their characterization of the effect of CXCR3 in fibrosis by using pan-CAF markers (eg podoplanin) and markers for other CAF populations eg inflammatory CAFs (iCAFs). Is collagen deposition also increased upon CXCR3 inhibition?

Reviewer #3 (Recommendations for the authors):

In this manuscript by Pandey and colleagues, the authors investigate regulation of macrophage phenotype during pancreatic cancer initiation. While prior studies have demonstrated significant pro- and anti-tumorigenic roles for macrophages during pancreatic tumorigenesis, mechanistic insights are mostly limited to those promoting pro-tumorigenic macrophage phenotypes, with little known about heterocellular interactions that promote a potentially tumoricidal or tumor-suppressive macrophage state. The data presented here support the novel hypothesis that a CXCL10-CXCR3 axis promotes an inflammatory macrophage identity and suppresses tumor initiation. The work is significant in providing insights into tissue homeostatic mechanisms that limit tumor formation. However, the authors' claims that peritoneal macrophages are the relevant macrophage pool for the phenotypes reported and that CXCL10-CXCR3 signaling controls inflammatory macrophage abundance via chemoattraction are not convincingly supported by the results in the manuscript. A limited number of edits and additional analyses would be helpful in clarifying the authors' findings.

Specific comments:

1. While the authors convincingly demonstrate that the CXCL10/CXCR3 axis regulates macrophage phenotype in vitro and in vivo, the significance of chemoattraction for the reported phenotypes is not clear. Tissue-resident macrophages express CXCR3 and are known to play significant roles in pancreatic tumor progression, proliferate in situ during tumor progression, and contribute to fibrosis, consistent with a phenotypic switch in these tissue-resident cells upon CXCL10-CXCR3 inhibition as the underlying mechanism as opposed to recruitment of peritoneal macrophages. Either the chemoattraction and recruitment model should be addressed experimentally in vivo, or text should be edited to accommodate this alternative model.

2. Total F4/80+ cells should be quantified in the genetically engineered KC model treated with CXCR3 NAB (results in Figure 3D) to address the impact of the CXCL10/CXCR3 axis on total macrophage abundance in these tissues and assess, in the presence of a functional immune system, whether macrophage phenotype alone is altered or if total abundance is also impacted by CXCL10-CXCR3 inhibition.

3. Several key references are absent from the introduction and should be added and briefly discussed in the context of this study, including prior studies of CXCL10 in PDAC (Hirth, Kuner et al., Gastroenterology, 2020; Romero, Gallinger et al., Clinical Cancer Research, 2020) and of macrophages/macrophage heterogeneity in pancreatic tumor initiation (Zhang, Pasca di Magliano et al., Gut, 2017; Zhu, DeNardo et al., Immunity, 2017).

---

## [Author Response]

1. Please clarify the statistical methods used and provide information on replicates for all data.

In the revised version of our manuscript we have revised the paragraph on Quantification and Statistical Analysis in the Experimental Procedures section to provide more detailed information. In addition to this, in each figure legend, we now provide information on replicates, statistical analyses and p-value.

2. The reviewers all agreed that the CXCL10 data comprises the core of the findings, but though those findings could be strengthened with further experiments. The current eLife policy also allows limiting the claims if new experiments are not possible. Specifically, to support the claim that CXCL10/CXCR3 blockade enhances cancer development, one suggested additional experiment is to evaluate tumor formation in KC mice upon longer treatment with CXCR3 Nab. However, this experiment would take a long time and thus may not be feasible. In the absence of further support that CXCL10/CXCR3 blockade enhances cancer development, and alternative would be to focus your conclusions on the role of CXCL10 in promoting an inflammatory macrophage identity. This latter option likely would include adjusting the title of the manuscript which refers to a role in tumor development.

One reviewer suggested an additional experiment to evaluate tumor formation in KC mice upon longer treatment with a CXCR3-NAB. Although we agree with the reviewer that this would be an interesting experiment, we feel that conducting this experiment would take a long time and require excessive resources (large quantities of CXCR3 NAB) and thus is not feasible at this point. Instead, as suggested by the editors, we now focus our conclusions on the role of CXCL10/CXCR3 signaling in promoting an inflammatory macrophage identity and promoting occurrence of precancerous low-grade lesions. We also adjusted the title of the manuscript, which now reads:

“CXCL10/CXCR3 signaling contributes to an inflammatory microenvironment and its blockade enhances progression of pancreatic precancerous lesions”.

3. Please address the questions raised by reviewers around the source of the macrophages as this speaks to the central claims of the study.

We agree with the editors and reviewers that this is an important question, with tissue resident macrophages, bone marrow-derived macrophages or peritoneal macrophages as potential responders to CXCL10.

With respect to early events leading to development of PDA, the influx of macrophages into the pancreas has been demonstrated following injury and during development and progression of pancreatic lesions [1,2]. Moreover, CXCL10 has been demonstrated as a chemoattractant for macrophages along with other immune cells [3]. Transwell invasion assays using both peritoneal and bone marrow-derived macrophages (M0, M1, M2) suggest that CXCL10 can act as a chemoattractant for M1-polarized macrophages (Figure 2B; Figure 2—figure supplement 1A).

However, since tissue resident macrophages have been attributed important roles in established pancreatic cancer (Zhu et al., 2017), we also determined if this population can be the recipients for CXCL10. Approximately 80% of tissue resident macrophages in normal mouse pancreas express CXCR3 (Figure 2—figure supplement 1B), but these cells when isolated do not proliferate in response to CXCL10 (Figure 2-figure supplement 1C). In sum, our in vitro data suggests that CXCL10 may drive the chemoattraction of inflammatory macrophages to the pancreas.

References for this response:

1. Gea-Sorli, S. and D. Closa, in vitro, but not in vivo, reversibility of peritoneal macrophages activation during experimental acute pancreatitis. BMC Immunol, 2009. 10: p. 42.

2. Liou, G.Y., et al., Mutant KRAS-induced expression of ICAM-1 in pancreatic acinar cells causes attraction of macrophages to expedite the formation of precancerous lesions. Cancer Discov, 2015. (1): p. 52-63.

3. Liu, M., S. Guo, and J.K. Stiles, The emerging role of CXCL10 in cancer (Review). Oncol Lett, 2011. 2(4): p. 583-589.

4. Address overstatements suggested by the reviewers.

We addressed all these points in the revised version of our manuscript (see responses to each reviewer below).

5. Some reviewers questioned how much the IRF4/5 data added to the paper? One possibility is to focus entirely on the role of CXCL10 in promoting an inflammatory macrophage identity, others thought this was interesting but perhaps is better included as supplementary data.

The data on IRF4/5 is based on previous published work in which it was shown that the balance between these two transcription factors determines if macrophages are M1 or M2 polarized [1, 2]. M2 macrophages mainly express IRF4, which is a key factor for their polarization phenotype [3]. Depletion of IRF4 in M2 macrophages results in a polarization change towards M1 macrophages [1]. Therefore, we feel, that the CXCR3-induced changes in IRF4 and IRF5 expression in M1 macrophages are a valid explanation for their change towards M2 polarization. But we also agree with the reviewers (and the editors) that this data is better included in the Supplemental Data (now Figure 3—figure supplement 1F).

References for this response:

1. Bastea, L.I., et al., Pomalidomide Alters Pancreatic Macrophage Populations to Generate an Immune-Responsive Environment at Precancerous and Cancerous Lesions. Cancer Res, 2019. 79(7): p. 1535-1548.

2. Gunthner, R. and H.J. Anders, Interferon-regulatory factors determine macrophage phenotype polarization. Mediators Inflamm, 2013. 2013: p. 731023.

3. Satoh, T., et al., The Jmjd3-Irf4 axis regulates M2 macrophage polarization and host responses against helminth infection. Nat Immunol, 2010. 11(10): p. 936-44.

Reviewer #1 (Recommendations for the authors):

General assessment of the work:

Overall, the manuscript addresses one piece of a major question in the pancreatic cancer field – how the inflammatory microenvironment of these tumors is sculpted through paracrine signaling pathways. In general, while the data reported are supportive of their conclusions, in most cases alternative explanations were not ruled out. This could be addressed with additional experiments. In particular, the question of the initial trigger for the CXCL10/CXCR3 pathway did not exclude other explanations and this was a major component of the story. In addition, there were a number of technical concerns and questions that limit our ability to interpret the data. These could be remedied in a resubmission but require careful attention to detail.Substantive concerns:1. Utilized ISH data as proof for protein expression. While necessary in order to identify which cell synthesize secreted proteins like CXCL10, it would be informative to see paired protein staining or Western blots for those experiments that don't already include them, as mRNA expression isn't always indicative of protein expression/levels.

In our opinion ISH is the method of choice to determine the cells that produce proteins that are secreted. In our experience simple IHC analyses does not provide reliable information if the protein of interest is produced by a cell or if it is bound to its target cell. The alternative, to perform Western blot analyses for protein expression from tissue would require precise microdissection, or the analyses of primary cells (i.e. SM3 cells for PanIN1), which are representative of the lesion that is to be analyzed. Whenever possible, we performed such analyses. However, we would like to point out that antibodies used for IHC need to be rigorously tested for non-specific signaling. For detection of IFNγ, for example, the antibodies we have tested for IHC (e.g. Bioss #0480R) failed our test.

2. In figure 1, the authors conclude that IFNγ is responsible for activation fo CXCL10 expression. Treatment with IFNγ did induce CXCL10, but this only proves that the pathway is sufficient and does not address whether it is necessary. To learn this, one could use an anti-IFNγ antibody and see if this blocks CXCL10 activation. Otherwise, the conclusions drawn from this experiment should be stated in a more constrained manner.

We would like to point out that there is a whole body of published work showing that IFNγ can induce CXCL10. CXCL10 is widely known in the literature as the interferon γ-inducible protein 10 (IP-10) and previously has been shown to be induced by IFNγ via activation of signal transducer and activator of transcription 1 (STAT1). Our data confirms this for SM3 PanIN cells. However, as suggested by the reviewer, in the revised version of our manuscript, we state the conclusions drawn from this set of experiments in a more constrained manner in the Results and Discussion sections.

3. Similarly, the conclusions drawn regarding T cell and NK cell involvement in the CXCL10/CXCR3 pathway are not fully supported.– The combined ISH/IF experiment in Figure 1G is an elegant technical approach to studying the source-cell of a secreted protein. However, not a single control was presented for this experiment.

This experiment has neighboring negative cells included as internal controls. For example, we show CD8+ cells that express IFNγ and neighboring CD8- cells that do not express IFNγ.

– Presence of T cells and NK cells in the area around precursor cells, and the knowledge that T cells and NK cells express IFNγ is not sufficient to conclude that T cells and NK cells are responsible for initiation of CXCL10 expression. It is a potential explanation, but no work was done to rule out other unknown factors.

We absolutely agree with the reviewer on this point and since presence of IFNγ in the pancreatic microenvironment is well established, there may be more cell types besides T cells and NK cells that produce IFNγ. Since the focus of our study was not to delineate/establish all the IFNγ producers in the tumor microenvironment, following the guidance of this reviewer, we have modified the text of our manuscript and suggest that T cells or NK cells could be a potential source of IFNγ in our experimental system. We also changed the model in Figure 5F and added “or other sources in the pancreas microenvironment”.

– Additionally, when done in a T cell free environment, the pathway still functioned and the authors themselves claim that there is "no significant role of T cells in CXCL10/CXCR3 driven development of pancreatic cancer." IFNγ from T cells cannot be both the initial driver and also not significant.

We agree with the reviewer that T cells cannot be both, “the initial driver” and also “not significant” at the same time. In a T cell free environment effects can be explained by presence of other IFNγ producing sells (such as NK cells) in the microenvironment.

4. In Figure 2B, was the migration assay performed on M2 macrophages? One would expect them not to be responsive based on 2A, but this should be examined.

We agree with the reviewer that this is an important control. In the revised manuscript (New Figure 2B) we have included a migration assay performed on M2 macrophages. We found that migration of M2-polarized macrophages is not affected by CXCL10. Since this macrophage population does not express CXCR3 (See Figure 2A) this is an expected result.

5. The conclusion that macrophage polarization occurred through the IRF4/5 transition is not sufficiently supported. It is not clear that the reported observation should even be included is it is not integral to the overall message of the paper.

This is based on previous published work in which it was shown that the balance between these two transcription factors determines if macrophages are M1 or M2 polarized [1, 2]. M2 macrophages express IRF4, which is a key factor for their polarization phenotype [3]. Depletion of IRF4 in M2 macrophages results in a polarization change towards M1 macrophages [1]. Therefore, we feel, that the CXCR3-induced changes in IRF4 and IRF5 expression in M1 macrophages are a valid explanation for their change towards M2 polarization. But we also agree with the reviewer (and the editors) that this data is better included in the Supplemental Data (now Figure 3—figure supplement 1D).

References for this response:

1. Bastea, L.I., et al., Pomalidomide Alters Pancreatic Macrophage Populations to Generate an Immune-Responsive Environment at Precancerous and Cancerous Lesions. Cancer Res, 2019. 79(7): p. 1535-1548.

2. Gunthner, R. and H.J. Anders, Interferon-regulatory factors determine macrophage phenotype polarization. Mediators Inflamm, 2013. 2013: p. 731023.

3. Satoh, T., et al., The Jmjd3-Irf4 axis regulates M2 macrophage polarization and host responses against helminth infection. Nat Immunol, 2010. 11(10): p. 936-44.

6. Please provide a statistical analysis of the data in 4G, particularly with regard to the incidence of higher grade lesions. Given the variance in the extent of PanIN lesions that present in the KC model, it is generally necessary to analyze far more than 4 animals per group to assess impact on this type of phenotype.

We absolutely agree with the reviewer that to obtain significant data on progression towards tumors when CXCR3 is neutralized would need not only more mice per experimental group, but also additional animal experiments where treatment occurs over a longer time span. We feel that this is beyond the scope of our manuscript. Therefore, we have removed the following paragraph from page 11:

“However, we found very few PanIN2 and PanIN3 lesions in mice treated with CXCR3 NAB (Figure 4G), but not in mice that were control treated. Although this has to be rigorously tested in futures studies, the occurrence of these high-grade lesions indicates that CXCL10/CXCR3 signaling may act protective for low grade lesions, and when neutralized may promote the formation of pancreatic cancer”.

We also changed Figure 4G and summarized further progressed lesions as “other”.

7. In general, for many experiments it was not clear how many replicates were performed and whether these were technical or biological replicates. It is important that this be stated the legend for each panel in each figure, otherwise it is impossible to provide an informed analysis of the data.

We absolutely agree with the reviewer that this information is essential for data analysis. All replicates mentioned in the manuscript refer to biological replicates. In the revision of our manuscript, this is clearly stated for each figure in the figure legend.

8. Related to the prior point, the data representation in most figures utilized outdated approaches (eg. bar plots) that communicate a minimum amount of information about distribution, variance, and replicates. We strongly recommend the adoption of more modern, informative data display conventions, such as those described in PMID: 32346721.

We thank the reviewer for this excellent point. in the current revision, we re-plotted all the bar plots and now provide 38 updated plots that communicate information about distribution, variance, and replicates.

9. Additionally, it is important to indicate which statistical test was utilized in each figure. The blanket statement in the Methods that Student's T was used is not sufficient as this is not always the appropriate test. For example, in Figure 4B, the authors present normalized ratiometric data (% positive area). A Student's T is not an appropriate test for this comparison because ratiometric data violate the assumption of a normal distribution required by this test. We strongly suggest the authors review every experiment with a biostatistician.

We absolutely agree with the reviewer and have worked with our biostatistician Dr. Yan Asmann (Associate Professor of BioMedical Informatics, Mayo Clinic) on the statistical analyses of all experiments. For example, for all experiments in which pancreatic areas were compared (previous Figures 4F and 5B), since % values are not normally distributed, we transformed the data via arcsin transformation before a t-test was performed. The statistical analysis used for each subfigure is now included in the revised figure legends.

Reviewer #2 (Recommendations for the authors):

CXCL10/CXCR3 contributes to an inflammatory microenvironment and its blockade enhances pancreatic cancer development

In this manuscript, Pandey et al., reports a novel CXCL10/CXCR3 signaling axis between pancreatic preneoplastic lesions and inflammatory macrophages to shape the microenvironment of PanIN. The authors provide strong in vivo experimental data to support a causal role of CXCL10 and CXCR3 in the recruitment and polarization of macrophages in the pancreas, as well as the degree of fibrosis in PanIN lesions. Addressing the following concerns will further strengthen the manuscript:

1. CXCL10 has previously been reported to be the major chemokine produced by pancreatic stellate cells (PSCs) in the PDA tumor microenvironment. CAFs expressing IP10 are thought to recruit Tregs to promote immune evasion and tumorigenesis. CXCL10 levels have also been shown to positively correlate with stromal content in human PDAC (PMID25415223). In the current manuscript, the authors report a very robust increase in fibrosis in PanIN (using SMA as a marker) when the CXCL10/CXCR3 axis is supressed using neutralizing antibodies. This raises the interesting question of whether CXCL10 exhibit opposite functions during premalignant vs malignant stages of tumorigenesis. This can be addressed using a syngeneic orthotopic model of PDA to evaluate the effect of CXCL10/CXCR3 inhibition in established tumors. Alternatively, the authors may consider revising the title of their manuscript to focus on the effect of CXCL10/CXCR3 in preneoplastic lesions as opposed to pancreatic cancer. CXCR3 Nab did not lead to a significant change in % of PanIN or ADM in mice (Figure 4G) so it is hard to extrapolate from the current data what the role of this signaling axis is in the context of cancer unless additional data is provided beyond the current timepoints to evaluate tumorigenesis.

We agree with the reviewer that investigating whether CXCL10 exhibits opposite functions during premalignant versus malignant stages of tumorigenesis is an exciting question. However, we feel that animal experiments to address this question rigorously are beyond the scope of the current manuscript. Therefore, we followed the reviewer’s suggestion and revised the title of our manuscript to focus on the effect of CXCL10/CXCR3 in preneoplastic lesions as opposed to pancreatic cancer. The new title is:

“CXCL10/CXCR3 signaling contributes to an inflammatory microenvironment and its blockade enhances progression of pancreatic precancerous lesions”.

In the revised manuscript we also addressed the possible opposite functions of CXCL10 in pre-malignant versus malignant stages in the last paragraph of the Discussion section on page 17. It reads:

“Considering our results, use of agonists for the receptor CXCR3 at stages of low-grade lesions may be useful to modulate macrophage polarization in the microenvironment such that a predominantly inflammatory population (M1) can be sustained. However, it needs to be noted that in pancreatic cancer expression of both CXCL10 and CXCR3 in tumor tissue has been correlated with poor prognosis (Liu et al., 2011; Lunardi et al., 2014). Therefore, it is unclear if treatment of pancreatic tumors with a CXCR3 agonist will result in a polarization switch of tumor associated macrophages that renders the lesion microenvironment less supportive for tumors, and increases efficiency of chemotherapy, or if it has a tumor promoting effect. This will be addressed in future studies”

2. What is the spatial relationship between interferon-γ-producing T cells and PanIN lesions? A multiplex FISH of CXCL10 and IFNγ will strengthen the model that short-distance-acting IFNγ from T cells within the pancreatic parenchyma induces CXCL10 production in neighbouring acinar/ADM cells. Along the same lines, what is the percentage of T cells in PanIN and ADM-bearing pancreata?

We agree with the reviewer that quantifying the T cell populations that produce IFNγ at ADM or PanIN lesions, as well as measuring their spatial proximity to these lesions is interesting, but we feel that this does not really belong into the current manuscript and requires a thorough analysis, which will be a separate project. However, we would like to point out the following to answer above reviewer questions: Multiple cell types within the pancreatic microenvironment, including T cells and NK cells, have been shown to secrete IFNγ [1-3]; and in Figure 1G we show that all of these are present and produce IFNγ. With respect to T cells, we (and others) previously have shown that they occur in proximity to ADM/PanIN areas in the KC model. For example, in Liou et al., Cell Reports, 2017 [4], we show in Supplemental Figure S3A that CD3- and CD4-positive cells are present in ADM/PanIN areas. Similarly, in Liou et al., Cancer Discovery, 2015 [5], we show in Supplemental Figure S1B, panel B7, that CD3-positive cells are present in ADM/PanIN areas. Moreover, in a more recent paper from Bastea et al., Cancer Research, 2019 [6], in Figures 6H and 6I we demonstrate the presence of CD3+;CD4+;IFNγ+ and CD3+;CD8+;IFNγ+ T cell populations in the pancreas of KC mice. An additional analysis of this data suggests that the percentage for CD3+;CD4+;IFNγ+ cells in ADM/PanIN bearing pancerata (n=3) is approximately 0.33% -/+ 0.28% of all cells and for CD3+;CD8+;IFNγ+ cells in ADM/PanIN bearing pancreata (n=3) is approximately 0.02% -/+ 0.02% of all cells.

References for this response:

1. Brauner, H., et al., Distinct phenotype and function of NK cells in the pancreas of nonobese diabetic mice. J Immunol, 2010. 184(5): p. 2272-80.

2. Castro, F., et al., Interferon-Γ at the Crossroads of Tumor Immune Surveillance or Evasion. Front Immunol, 2018. 9: p. 847.

3. Corthay, A., et al., Primary antitumor immune response mediated by CD4+ T cells. Immunity, 2005. 22(3): p. 371-83.

4. Liou, G.Y., et al., The Presence of Interleukin-13 at Pancreatic ADM/PanIN Lesions Alters Macrophage Populations and Mediates Pancreatic Tumorigenesis. Cell Rep, 2017. 19(7): p. 1322-1333.

5. Liou, G.Y., et al., Mutant KRAS-induced expression of ICAM-1 in pancreatic acinar cells causes attraction of macrophages to expedite the formation of precancerous lesions. Cancer Discov, 2015. 5(1): p. 52-63.

6. Bastea, L.I., et al., Pomalidomide Alters Pancreatic Macrophage Populations to Generate an Immune-Responsive Environment at Precancerous and Cancerous Lesions. Cancer Res, 2019. 79(7): p. 1535-1548.

3. Tumor associated macrophages are generally thought to originate from circulating monocytes derived from bone marrow HSCs. While thioglycollate activated peritoneal macrophages is a tractable experimental model, they are phenotypically very different from non-elicited, resident peritoneal macrophages, or naïve bone marrow derived macrophages, and as such do not serve as a proper control for "non-polarized" macrophages. Experiments presented in Figure 2A and B should also be done using naïve (M0) and M1 , M2 macrophages derived from the bone marrow. Experimental evidence showing that CXCR3-expressing macrophages in the pancreas originated from the peritoneum is currently absent, the authors should therefore refrain from drawing conclusions on the origin of these macrophages.

This is an excellent point. For the revised version of our manuscript we performed additional experiments using naïve and M1-polarized or M2-polarized macrophages derived from the bone marrow, and obtained data (shown in New Figure 2—figure supplement 1A) consistent with previous data obtained with polarized peritoneal macrophages. We also agree that there is no experimental evidence of the origin of the chemoattracted macrophages and made appropriate changes throughout the text to refrain from implicating that they originate in the peritoneum.

4. The authors should extend their characterization of the effect of CXCR3 in fibrosis by using pan-CAF markers (eg podoplanin) and markers for other CAF populations eg inflammatory CAFs (iCAFs). Is collagen deposition also increased upon CXCR3 inhibition?

We agree with the reviewer that determining effects on different CAF populations will be an interesting question. However, the major point in our manuscript (Figure 4) is to determine the effects of CXCR3 signaling on alternatively-activated (M2) macrophage populations. We have included an analysis of SMA+ fibroblasts (see Figures 4A and 4B) because we previously have shown that in the KC model M2 macrophages (marked by expression of Ym1) regulate this population, and a depletion of M2 macrophages decreases fibrosis as measured by staining for SMA [1]. This is why we used IHC for SMA as readout for fibrosis affected by the M2 macrophage population. A detailed analysis of effects of CXCR3 signaling on different CAF populations is clearly beyond the scope of the current study since there are no single markers for these populations described. Such analyses need co-staining for several markers (i.e. IF-IHC for podoplanin and SMA; or IHC for podoplanin combined with ISH for CXCL12) to distinguish myCAFs from non-myCAFs or to determine changes in iCAFs by analyzing CD45-/Ly6c+ population changes. We feel that this is a project by itself and will be addressed in future studies.

References for this response:

1. Liou, G.Y., et al., The Presence of Interleukin-13 at Pancreatic ADM/PanIN Lesions Alters Macrophage Populations and Mediates Pancreatic Tumorigenesis. Cell Rep, 2017. 19(7): p. 1322-1333.

Reviewer #3 (Recommendations for the authors):

In this manuscript by Pandey and colleagues, the authors investigate regulation of macrophage phenotype during pancreatic cancer initiation. While prior studies have demonstrated significant pro- and anti-tumorigenic roles for macrophages during pancreatic tumorigenesis, mechanistic insights are mostly limited to those promoting pro-tumorigenic macrophage phenotypes, with little known about heterocellular interactions that promote a potentially tumoricidal or tumor-suppressive macrophage state. The data presented here support the novel hypothesis that a CXCL10-CXCR3 axis promotes an inflammatory macrophage identity and suppresses tumor initiation. The work is significant in providing insights into tissue homeostatic mechanisms that limit tumor formation. However, the authors' claims that peritoneal macrophages are the relevant macrophage pool for the phenotypes reported and that CXCL10-CXCR3 signaling controls inflammatory macrophage abundance via chemoattraction are not convincingly supported by the results in the manuscript. A limited number of edits and additional analyses would be helpful in clarifying the authors' findings.

Specific comments:

1. While the authors convincingly demonstrate that the CXCL10/CXCR3 axis regulates macrophage phenotype in vitro and in vivo, the significance of chemoattraction for the reported phenotypes is not clear. Tissue-resident macrophages express CXCR3 and are known to play significant roles in pancreatic tumor progression, proliferate in situ during tumor progression, and contribute to fibrosis, consistent with a phenotypic switch in these tissue-resident cells upon CXCL10-CXCR3 inhibition as the underlying mechanism as opposed to recruitment of peritoneal macrophages. Either the chemoattraction and recruitment model should be addressed experimentally in vivo, or text should be edited to accommodate this alternative model.

In normal pancreata of 7-8 weeks old mice we detect only very few tissue resident macrophages, with an average of 5.51 -/+ 4.06 macrophages per mm2. Of those approximately 80% were also CXCR3+ (New Figure 2—figure supplement 1B). However, tissue resident macrophages after isolation and stimulation with CXCL10 did not show altered proliferation (New Figure 2—figure supplement 1C).

For the KC model, we feel that we have shown in previous papers that inflammatory macrophages can be attracted to the pancreas and contribute to development of early lesions (e.g. Liou et al., Cancer Discovery, 2015; [1]). However, we also agree with the reviewers that it is unclear if these originate from the bone marrow or the peritoneum. Here, for CXCL10, we show that it attracts both BMD and peritoneal macrophages (Figure 2B and New Figure 2—figure supplement 1A).

In the revised version of our manuscript, we follow the reviewer’s guidance and have edited the text to discuss the potential source of macrophages in more detail. The new text is included on pages 7/8 and reads: “Transwell invasion assays using both, peritoneal and bone marrow-derived macrophages suggest that CXCL10 can act as a chemoattractant for M1-polarized macrophages (Figure 2B; Figure 2—figure supplement 1A). However, since tissue resident macrophages have been attributed important roles in established pancreatic cancer (Zhu et al., 2017), we also determined if this population can be the recipients for CXCL10. Approximately 80% of tissue resident macrophages in normal mouse pancreas express CXCR3 (Figure 2—figure supplement 1A), but these cells, when isolated do not proliferate in response to CXCL10 (Figure 2—figure supplement 1B). In sum our in vitro data suggests that CXCL10 may drive the chemoattraction of inflammatory macrophages to the pancreas”.

References for this response:

1. Liou, G.Y., et al., Mutant KRAS-induced expression of ICAM-1 in pancreatic acinar cells causes attraction of macrophages to expedite the formation of precancerous lesions. Cancer Discov, 2015. 5(1): p. 52-63.

2. Total F4/80+ cells should be quantified in the genetically engineered KC model treated with CXCR3 NAB (results in Figure 3D) to address the impact of the CXCL10/CXCR3 axis on total macrophage abundance in these tissues and assess, in the presence of a functional immune system, whether macrophage phenotype alone is altered or if total abundance is also impacted by CXCL10-CXCR3 inhibition.

This is an excellent point and we have included a quantification of total F4/80+ cells in the New Figure 3—figure supplement 1F.

3. Several key references are absent from the introduction and should be added and briefly discussed in the context of this study, including prior studies of CXCL10 in PDAC (Hirth, Kuner et al., Gastroenterology, 2020; Romero, Gallinger et al., Clinical Cancer Research, 2020) and of macrophages/macrophage heterogeneity in pancreatic tumor initiation (Zhang, Pasca di Magliano et al., Gut, 2017; Zhu, DeNardo et al., Immunity, 2017).

We thank the reviewer for pointing out these references and apologize for having missed them in our initial manuscript. In our revision these publications are now included.